# Learning With Multi-Group Guarantees For Clusterable Subpopulations

Jessica Dai [1]   Nika Haghtalab [1]   Eric Zhao [1]

## Abstract

A canonical desideratum for prediction problems is that performance guarantees should hold not just on average over the population, but also for meaningful subpopulations within the overall population. But what constitutes a meaningful subpopulation? In this work, we take the perspective that relevant subpopulations should be defined with respect to the clusters that naturally emerge from the distribution of individuals for which predictions are being made. In this view, a population refers to a mixture model whose components constitute the relevant subpopulations. We suggest two formalisms for capturing per-subgroup guarantees: first, by attributing each individual to the component from which they were most likely drawn, given their features; and second, by attributing each individual to all components in proportion to their relative likelihood of having been drawn from each component. Using online calibration as a case study, we study a multi-objective algorithm that provides guarantees for each of these formalisms by handling all plausible underlying subpopulation structures simultaneously, and achieve an $O(T^{1/2})$ rate even when the subpopulations are not well-separated. In comparison, the more natural *cluster-then-predict* approach that first recovers the structure of the subpopulations and then makes predictions suffers from a $O(T^{2/3})$ rate and requires the subpopulations to be separable. Along the way, we prove that providing per-subgroup calibration guarantees for underlying clusters can be easier than learning the clusters: separation between median subgroup features is required for the latter but not the former.

---

[*]Alphabetical order. [1]Department of Electrial Engineering and Computer Science, U.C. Berkeley, Berkeley, CA, USA. Correspondence to: Jessica Dai <jessicadai@berkeley.edu>.

*Proceedings of the $42^{nd}$ International Conference on Machine Learning*, Vancouver, Canada. PMLR 267, 2025. Copyright 2025 by the author(s).

## 1. Introduction

For systems that make predictions about individuals, it is well-understood that good performance on average across the population may not imply good performance at an individual level. On the other hand, while the ideal system might be one that can provide per-individual performance guarantees, such a system may be intractable to learn from data, if it exists at all. To address these challenges, *per-subpopulation* guarantees have emerged as a widely-accepted approach that balances tractability with ensuring good performance across subpopulations (e.g., Blum et al. (2017); Hébert-Johnson et al. (2018); Hashimoto et al. (2018); Lahoti et al. (2020); Wang et al. (2020); Haghtalab et al. (2022)). Such guarantees may also be desirable for normative or regulatory reasons to capture notions of fairness, or because domain shift often involves changes in the proportions of subgroups. Therefore, the subpopulations for which guarantees are provided should be those that are deemed especially significant, salient, or relevant.

What, then, defines a relevant subpopulation? One influential perspective considers a subpopulation as a predefined combination of feature values, where individuals are represented as feature vectors (e.g., Hébert-Johnson et al. (2018)). In our work, we take an alternative view on what constitutes a subpopulation of interest. We propose that the relevant subgroups for a particular prediction task should be exactly those subgroups that emerge endogenously within the distribution of the individuals being considered for that task. This means that the group membership(s) of any individual cannot be determined through their features alone; instead, their group identity can be understood only by placing their individual features in the context of the rest of the population. In effect, rather than being defined *a priori*, these subgroups must be *learned* about from data in an unsupervised sense.

### 1.1. Defining subpopulations via statistical identifiability

One reason to think of group membership in context of the population rather than as deterministic functions of individual features is a normative perspective. A common critique of standard practice in modeling subgroups is that observable (demographic) features are only approximations of more complex phenomena that are related to—but not directly causal of—shared life experience. Therefore, de-

manding "equal performance" across rigid (demographic) categories does not necessarily imply "fairness" in a normative sense (see, e.g., Benthall & Haynes (2019); Hu & Kohler-Hausmann (2020); Hu (2023) for more extended discussion). In some sense, our approach can be seen as an attempt to develop a more constructivist perspective on defining subpopulations—placing individuals in context with others for whom those predictions are made, and allowing group definitions to vary based on the particular prediction task—as opposed to an essentialist one. Of course, we cannot claim to fully resolve these normative challenges or realize these goals; however, we think of them as a reason to explore different ways of understanding the relationship between groups and individuals.

Because we cannot determine group membership solely based on an individual's feature vector, our problem setting requires some structure on the domain; in particular, features must be clusterable. Then, if all one initially knows about the population is that it is comprised of multiple subpopulations where group membership affects feature realizations, determining subgroup membership based only on those realizations is the best one can expect to do. Our focus on statistically identifiable groups is in contrast with the computationally-identifiable groups studied when subpopulations are defined as combinations of feature values. In those settings, it is necessary to ensure that membership can be distinguished as *efficiently* as possible (e.g., through low circuit complexity, as multicalibration was initially described in Hébert-Johnson et al. (2018)); in our setting, the key challenge is to instead identify membership as *accurately* as possible, because group membership itself is uncertain.

We also note that finding statistically identifiable subpopulations from data (in the sense of learning membership likelihoods), and using those subgroups downstream, is common in audit settings when true subgroup labels are unknown. In these cases, inferring or estimating group membership is a natural (and sometimes even necessary) approach. For example, it is well-known that names are often associated with demographic identity (Elder & Hayes, 2023), and audits of resume screening systems in practice often use those assumed associations rather than explicitly-stated demographic identity (e.g., Kang et al. (2016); Wilson & Caliskan (2024)). More generally, an extensive literature discusses how demographic labels might be imputed from data—e.g., name and census tract, in the well-known "BISG" (Bayesian Improved Surname Geocoding) approach (Elliott et al., 2009) and its variants; how those labels might be used for downstream purposes (e.g. auditing lending decisions (Zhang, 2018)); and how those estimates ought to be incorporated in a mathematical sense to those downstream applications (e.g., Dong et al. (2024)).

## 1.2. Our approach

To operationalize our approach, our measures of per-group performance must handle the fact that any individual's group membership can be at best approximately inferred, e.g. as probabilities representing the likelihood that that individual belonged to each group. Accordingly, we study two natural approaches to measuring per-group error. The first, which we refer to as *discriminant error*, attributes the error an individual experiences only to the group that the individual *most likely* belongs to. This corresponds to typical notions of clustering error and is often used in existing approaches to handling uncertainty in group membership, which is effectively to ignore it (see, e.g., discussion in Dong et al. (2024)). We also study a probabilistic alternative, which we term *likelihood error*, where we attribute the error an individual experiences to every group, but weighted according to the likelihoods of membership in each group. This likelihood-based notion of per-group error explicitly acknowledges the existence of meaningful uncertainty in group membership. As a consequence, likelihood error also provides some reasonable robustness properties (e.g., to changes in the relative proportions of subgroups), and, relative to discriminant error, improves guarantees for subgroups that comprise a smaller proportion of the total population.

Both of these measures, however, require knowledge of the subgroup distributions—that is, the likelihood that any particular individual (feature vector) belongs to (was drawn from) any particular group—which depends on the population distribution for the prediction task, and are therefore initially unknown. The natural strategy for addressing the problems of unknown subgroups and unknown labels is to first complete the unsupervised task of learning the subgroups, then for each of the learned distributions complete the supervised task of learning to predict; we call this the *cluster-then-predict* approach. The overall prediction quality of this approach critically depends on how well the "clustering" stage can be performed—a task that often requires a large number of observations and separation between subpopulations. We circumvent this problem through a *multi-objective* approach (e.g., Haghtalab et al. (2022; 2023)) where, instead of learning the exact underlying clustering, we construct a class of plausible clusterings and provide high-quality per-group predictions for all of them simultaneously. What clusterings are "plausible," and how can we provide a solution that works for all of them? These constitute the central technical thrusts of our work.

**1) Understanding the structure of subpopulations.** We consider the class of all plausible group membership functions corresponding to the two formalisms of discriminant and likelihood error. We show that for both formalisms, the complexity of the appropriate class is characterized by the pseudodimension of possible likelihood ratios. Furthermore,

when our model is instantiated with a mixture of exponential families (e.g. Gaussian mixture models), we show that the pseudodimension of group membership functions is linearly bounded by the dimension of the sufficient statistic.

**2) Multi-objective algorithms for per-subgroup guarantees.** We leverage recent results in multi-objective learning to minimize error simultaneously over all possible clusterings of the data. Using calibration as a case study, we show how to extend recent online multicalibration algorithms (Haghtalab et al., 2023) so that they provide per-subpopulation guarantees for classes of real-valued distinguisher functions of low complexity, and by extension, families of clustering functions. For both discriminant and likelihood calibration error, our multi-objective approach achieves $O(T^{1/2})$ online error without requiring separability in the underlying clusters. This is in contrast to the cluster-then-predict approach, for which we demonstrate $O(T^{2/3})$ error rates even under separability assumptions.

**3) Towards statistically-identifiable subpopulations.** Beyond the technical approach, we view our work as an important step towards reasoning about group membership in context of the actual population on which predictions are being made. We argue that subpopulations should be defined endogenously, rather than characterized by explicit combinations of feature values. Our framework also provides a language for formalizing the relationship between explicitly learning subgroups, as opposed to providing high quality predictions for them: in fact, the former (which often requires separation between subpopulation means) is not necessary for the latter.

### 1.3. Related work

**Multicalibration.** Calibration is a well-studied objective in online forecasting (Dawid, 1982; Hart, 2022), with classical literature having studied calibration across multiple sub-populations (Foster & Kakade, 2006) and recent literature having studied calibration across computationally-identifiable feature groups (Hébert-Johnson et al., 2018). The latter thread of work, known as multicalibration, has found a wide range of connections to Bayes optimality, conformal predictions, and computational indistinguishability (Dwork et al., 2021; Gopalan et al., 2022; Gupta et al., 2022; Hébert-Johnson et al., 2018; Jung et al., 2021; 2023). We use online multicalibration algorithms (Gupta et al., 2022; Haghtalab et al., 2023) as a building block for efficiently obtaining per-group guarantees in our model.

**Fair machine learning.** The fair machine literature has developed various approaches to handling uncertainty in group membership. One strategy is to avoid enumerating subgroups entirely, and instead focus on identifying subsets of the domain where prediction error is high (Hashimoto et al., 2018; Lahoti et al., 2020). A separate line of work considers learning when demographic labels are available but noisy (Awasthi et al., 2020; Wang et al., 2020). Yet another approach is to use a separate estimator for group membership (Awasthi et al., 2021; Chen et al., 2019; Kallus et al., 2022); we note that our notions of per-group performance could be applicable to these methods as well, even if our definitions of "group" are different. Liu et al. (2023) also propose a means of understanding group identity in context with the rest of the population, in this case through social networks.

Finally, though intersectionality is not the focus of our work, our model of subpopulations is one approach to providing guarantees for unobservable marginalized subgroups. In particular, we see our model as an alternative to the practice of enumerating all (possibly-overlapping) subgroups defined by intersections of feature values, as in Hébert-Johnson et al. (2018), which assumes that unobservable marginalized subgroups can be approximated through such feature sets. We seek to to explicitly incorporate intrinsic structure in covariates across groups, as suggested in Wang et al. (2020); more generally, to the extent that power is central to what "defines" a group (Ovalle et al., 2023), our model gives an application-specific way to discuss it (i.e., in terms of how subgroup distributions appear).

## 2. Preliminaries

**Our generative model.** Let $\mathcal{X} \in \mathbb{R}^d$ denote a $d$-dimensional instance space and $\mathcal{Y} = \{0, 1\}$ denote the *label* or *outcome* space. We consider a generative model over $\mathcal{X} \times \mathcal{Y}$, where instances are generated from a mixture of $k$ distributions and the conditional outcome distribution is independent of the component from which the instance is generated. Formally, we define a discrete hidden-state *endogenous subgroups generative model* $f$, such that

$$f(x, y) \propto f(y \mid x)f(x \mid g)f(g),$$

where $f(g) = w_g$ is the distribution over $[k]$ corresponding to mixing weights $w_g \in [0, 1]$ with $\sum_{g \in [k]} w_g = 1$; $f(x \mid g)$ is the density of component $g$, and $f(y \mid x)$ is a conditional label distribution that is independent of $g$. In this work, we assume that $f(x \mid g)$ belongs to a class of densities $\mathcal{F}$. For example, $\mathcal{F}$ could indicate the class of Gaussian mixture models (as we study in Section 3), or an exponential family (as we discuss in Section 4). Then, each pair $(x, y)$ is generated by first sampling integer $g \in [k]$ according to weights $(w_1, \ldots, w_k)$; then sampling $x \sim f(x \mid g)$, and finally sampling $y$ according to $f(y \mid x)$. For clarity, we will often suppress $w_g$ in the following exposition, but our results follow without loss of generality as long as all $w_g$ are bounded below by a constant.

**Online prediction.** Our high level goal is to take high-quality actions for instances that are generated from an unknown endogenous subgroups model. Let $\mathcal{A}$ denote the action space. Examples of action spaces for prediction tasks include $\mathcal{A} = \{0,1\}$ where an action refers to a predicted label, or $\mathcal{A} = [0,1]$ where an action refers to predicting the probability that the label is 1. We consider an online prediction problem where a sequence of instance-outcome pairs $(x_1, y_1), \ldots, (x_T, y_T)$ is generated i.i.d. from an unknown generative model $f$ supported on $\mathcal{X} \times \mathcal{Y}$. At time $t$, the learner must take an irrevocable action $a_t \in \mathcal{A}$ having seen only $x_{1:t}$ and $y_{1:t-1}$. Equivalently, a learner can be thought of as choosing a function $p_t : \mathcal{X} \to \mathcal{A}$ that maps any feature $x$ to an action $a$. From this perspective, at time $t$ the learner chooses $p_t$ having only observed $x_{1:t-1}$ and $y_{1:t-1}$, after which $(x_t, y_t)$ is observed and the learner takes action $p_t(x_t)$.

**Performance on subpopulations.** We evaluate the quality of the learner's actions using a vector-valued loss function $\ell : \mathcal{A} \times \mathcal{Y} \to \mathcal{E}$ where $\mathcal{E}$ is some Euclidean space. Examples of loss functions include the scalar binary loss function $\ell(a, y) := \mathbb{1}[a \neq y]$ and the vector-based calibration loss $\ell(a, y)_v = \mathbb{1}[a = v](a - y)$.

In the style of Blackwell approachability (Blackwell, 1956), the learner's overall goal is to produce actions that lead to small cumulative loss on all the relevant subpopulations in the sequence $(x_1, y_1), \ldots, (x_T, y_T)$, as measured by a norm $\|\cdot\|$. We envision relevant subpopulations to be exactly the mixture components of our generative model. However, it is not possible to determine the component that generates an instance $(x, y)$ in a mixture model. Instead, we consider two notions of performing well on subpopulations. In the first, we purely attribute each $(x, y)$ to the component $g = \arg\max_{j \in [k]} f(j \mid x, y)$ that was most likely responsible for producing $(x, y)$; that is, we aim to minimize

$$\max_{g \in [k]} \left\| \sum_{t=1}^{T} \mathbb{1}\left[ g = \arg\max_{j \in [k]} f(j \mid x_t, y_t) \right] \cdot \ell(a_t, y_t) \right\|.$$

The attribution of $(x, y)$ to the most likely component corresponds to the usual task of clustering as is done in practice.

In the second, we attribute each $(x, y)$ to a subpopulation $g$ with probability $f(g \mid x, y)$; that is, we aim to minimize

$$\max_{g \in [k]} \left\| \sum_{t=1}^{T} f(g \mid x_t, y_t) \cdot \ell(a_t, y_t) \right\|.$$

Note that by definition, $f(g \mid x, y) = \frac{f(x \mid g) w_g}{\sum_{j \in [k]} f(x \mid j) w_j}$ is the probability $g$ was indeed responsible for producing $(x, y)$. This objective considers the contribution of an individual to subgroup error in proportion to the uncertainty of that

individual's "group membership." Additionally, note that this objective is robust to reweightings of subpopulations, as $\mathbb{E}\left[f(g \mid x_t, y_t) \ell(a_t, y_t)\right] = f(g) \mathbb{E}\left[\ell(a_t, y_t) \mid g\right]$.

## 2.1. Model instantiation for calibration loss

For concreteness, this paper instantiates our model for the task of producing predictions that are *calibrated* with respect to clusterable subpopulations. However, our endogenous subgroups model extends beyond calibration to a number of other settings that can be studied under online approachability, such as online *conformal prediction* and *calibeating* (Jung et al., 2023; Lee et al., 2022).

For studying calibrated predictions, we work with action space $\mathcal{A} = [0, 1]$ and let $\widehat{y} \in \mathcal{A}$ correspond to the predicted probability that $y = 1$. Calibration is a common requirement on predictors, necessitating their predictions to be unbiased conditioned on the predicted value. For technical reasons such as dealing with the fact that predicted values can take any real values, calibration is more conveniently defined by considering buckets of predicted values. Formally, we define a set of buckets $V_\lambda = \{0, 1/\lambda, 2/\lambda, \ldots, 1\}$ and say that prediction $\widehat{y}$ belongs to bucket $v$ (denoted by $\widehat{y} \in v$) when $|\widehat{y} - v| \leq 1/2\lambda$. Then, the calibration error of a sequence of predictors $p_1, \ldots, p_T$ on instance-outcome pairs $(x_1, y_1), \ldots, (x_T, y_T)$ is defined by $\max_{v \in V_\lambda} \left| \sum_{t=1}^{T} \mathbb{1}\left[p_t(x_t) \in v\right] \cdot (p_t(x_t) - y_t)\right|$. In other words, calibration loss is the cumulative $\ell_\infty$ norm of the objective $\ell(a, y) = [\mathbb{1}[a \in v] \cdot (a - y)]_{v \in V_\lambda}$.

We can accordingly define two variants of the calibration error that account for miscalibration as experienced by each component. In these definitions, we take $\lambda > 0$ to be fixed and clear from the context and suppress it in the notations.

In our first definition, called the *discriminant calibration error*, an instance $(x, y)$ is purely attributed to the component $g = \arg\max_{j \in [k]} f(j \mid x, y)$ that was most likely responsible for producing $(x, y)$.

**Definition 2.1** (Discriminant Calibration Error)**.** Given a sequence of instance-outcome pairs $(x_1, y_1), \cdots, (x_T, y_T)$, the *discriminant calibration error* of predicted probabilities $\widehat{y}_1, \ldots, \widehat{y}_T$ with respect to the endogenous subgroups model $f$—as specified by distributions $f(y|x)$, $f(x|g)$, and $f(g)$—is defined as

$$\mathbf{DCE}_f(\widehat{y}_{1:T}, x_{1:T}, y_{1:T}) :=$$

$$\max_{g \in [k]} \max_{v \in V_\lambda} \left| \sum_{t=1}^{T} \mathbb{1}\left[ g = \arg\max_{j \in [k]} f(j \mid x_t, y_t) \right] \right.$$
$$\left. \cdot \mathbb{1}\left[\widehat{y}_t \in v\right] \cdot (\widehat{y}_t - y_t) \right|.$$

When predictors $p_1, \ldots, p_T$ are used for making predictions $\widehat{y}_t = p_t(x_t)$, we denote the corresponding discriminant calibration error by $\mathbf{DCE}_f(p_{1:T}, x_{1:T}, y_{1:T})$.

In our second approach, *likelihood calibration error*, $(x, y)$ is attributed to any component $g$ with likelihood $f(g \mid x, y)$.

**Definition 2.2** (Likelihood Calibration Error)**.** Given a sequence of instance-outcome pairs $(x_1, y_1), \cdots, (x_T, y_T)$, the *likelihood calibration error* of predicted probabilities $\widehat{y}_1, \ldots, \widehat{y}_T$ with respect to the endogenous subgroups model $f$—as specified by distributions $f(y|x)$, $f(x|g)$, and $f(g)$—is defined as

$$\mathbf{LCE}_f(\widehat{y}_{1:T}, x_{1:T}, y_{1:T}) :=$$
$$\max_{g \in [k]} \max_{v \in V_\lambda} \left| \sum_{t=1}^{T} f(g \mid x_t, y_t) \cdot \mathbb{1}\left[\widehat{y}_t \in v\right] \cdot (\widehat{y}_t - y_t) \right|.$$

When predictors $p_1, \ldots, p_T$ are used for making predictions $\widehat{y}_t = p_t(x_t)$, we denote the corresponding likelihood calibration error by $\mathbf{LCE}_f(p_{1:T}, x_{1:T}, y_{1:T})$.

## 2.2. Additional notation

Finally, we provide definitions for the following tools we will use in our analysis.

*Pseudodimension* is a generalization of the Vapnik-Chervonenkis (VC) dimension for real-valued functions.

**Definition 2.3** (Anthony & Bartlett (1999), Chapter 11)**.** Let $\mathcal{F} : x \mapsto \mathbb{R}$ be a set of real-valued functions. Then, the pseudodimension of $\mathcal{F}$, notated $\mathrm{Pdim}(\mathcal{F})$, is the VC-dimension of the class $\{\mathrm{sgn}(f(x) - y) : f \in \mathcal{F}, y \in \mathbb{R}\}$.

*Exponential families* are a class of statistical models where densities have a common structure (see, e.g., Fithian (2023)). Examples include Poisson, Beta, and Gaussian distributions.

**Definition 2.4.** A statistical model $\mathcal{F} = \{f_\theta : \theta \in \Theta\}$ is an exponential family if it can be defined by a family of densities of form $f_\theta(x) = h(x) \exp(\langle \theta, T(x) \rangle - A(\theta))$, where $\theta \in \Theta$ is the parameter that uniquely specifies each density. The quantity $T(x)$ is known as the sufficient statistic.

In particular, note that a multivariate Gaussian density with mean $\mu$ and covariance $\Sigma$ can be written as

$$f_{\mu, \Sigma}(x) = \frac{1}{(2\pi)^{d/2}|\Sigma|^{1/2}} \exp\left(-\frac{1}{2}(x - \mu)^\top \Sigma^{-1}(x - \mu)\right)$$
$$= \exp\left(\begin{bmatrix} \Sigma^{-1}\mu \\ -\frac{1}{2}\mathrm{vec}(\Sigma^{-1}) \end{bmatrix}^\top \begin{bmatrix} x \\ \mathrm{vec}(xx^\top) \end{bmatrix} - A(\mu, \Sigma)\right),$$

with $A(\mu, \Sigma) = \frac{1}{2}\mu^\top \Sigma^{-1}\mu + \frac{1}{2}\log|\Sigma| + \frac{d}{2}\log(2\pi)$, and $\dim(T(x)) = \dim\left(\binom{x}{\mathrm{vec}(xx^\top)}\right) = d(d+1)/2$.

## 3. A first attempt: Cluster-then-predict for Gaussian mixtures

As a warmup, we consider the natural algorithmic approach: to first spend some timesteps to estimate the underlying

group structure, and then provide guarantees for the estimated groups. We will focus on minimizing discriminant calibration error for a simplified problem setting, then highlight some challenges involved in extending the approach to (a) more general problem settings and (b) to minimizing likelihood calibration error.

**Minimizing discriminant calibration error via cluster-then-predict.** To instantiate cluster-then-predict for discriminant calibration error, we leverage two common types of algorithms. For the first phase, we use a clustering algorithm that, given a Gaussian mixture model, outputs a mapping $F : \mathcal{X} \to \{1, 2\}$ indicating the group memberships. In particular, we use the algorithm of Azizyan et al. (2013) to obtain $F$ for which $F(x) = \arg\max_{j \in \{1, 2\}} f(j \mid x)$ for all but an $\varepsilon$ fraction of the underlying distribution, after having made $O(1/\varepsilon^2)$ observations. For the second phase, we can instantiate one (online) calibration algorithm that provides marginal calibration guarantees for its predictions on each of the two clusters. For example, Foster & Vohra (1998) guarantees at most $T^{1/2}$ calibration error.[1]

---

Algorithm 1: *Cluster-Then-Predict Algorithm for Minimizing DCE*

For the first $T' < T$ timesteps, make arbitrary predictions and collect observed features $x_1, \ldots, x_{T'}$. Apply a clustering algorithm, such as the Azizyan et al. (2013) estimator, to the observed features to partition the domain into cluster assignments $F : \mathcal{X} \to \{1, 2\}$.

Then, instantiate two calibrated prediction algorithms (e.g., the Foster & Vohra (1998) algorithm), one for each cluster. For every subsequent timestep $t = T' + 1, \ldots, T$, observe $x_t$ and predict $\widehat{y}_t$ by applying a calibrated online forecasting algorithm to the transcript $\{(x_\tau, y_\tau) \mid T' < \tau < t, F(x_\tau) = F(x_t)\}$ consisting only of datapoints with the same predicted cluster assignment.

---

We formalize the guarantees of this approach in Prop. 3.1.

**Proposition 3.1.** *Let $f$ be an unknown endogenous subgroups model whose Gaussian components are isotropic with $\|\mu_1 - \mu_2\| \geq \gamma$. Then, with probability $1 - \delta$, the* Cluster-Then-Predict *algorithm attains discriminant calibration error of $O(d^{1/3}T^{2/3}\gamma^{-4/3} + \sqrt{T \log(1/\delta)})$, when it is run with an appropriate choice of $T' = \Theta(d^{1/3}T^{2/3}\gamma^{-4/3})$.*

---

[1]The guarantees of Foster & Vohra (1998) hold even when $x_t$ are adversarial; an even simpler algorithm would suffice for our i.i.d. case. For example, one could take the naive approach of using some additional timesteps to estimate $\mathbb{E}[y \mid g]$ to sufficient accuracy and use those estimated means for the remainder of time.

*Proof Sketch.* This $T^{2/3}$ rate is typical for two-stage online algorithms, such as explore-then-commit, and its proof is also similar. By Azizyan et al. (2013) (see Theorem B.1), learning a cluster assignment $F$ that has $\varepsilon$ error takes $T' = \Theta(d/\gamma\varepsilon^2)$ samples. Note that, the DCE of this algorithm is at most $T' + \varepsilon T + \sqrt{T \ln(1/\delta)}$, where the second term accounts for the clustering mistakes and the third term accounts for the calibration error of the "predict" stage. Setting $T' = \Theta(d^{1/3}T^{2/3}\gamma^{-4/3})$ gives the desired bound. $\square$

*Remark* 3.2. In addition to the upper bound, the result of Azizyan et al. (2013) also implies that, for this class of cluster-then-predict algorithms, the $O(T^{2/3})$ rate is in fact minimax optimal: it is impossible to learn $F$ to any higher accuracy with any fewer samples.

*Remark* 3.3. Another consequence of the cluster-then-predict approach is that any hardness in learning cluster membership is inherited. For example, the $\gamma$-separation dependence of Proposition 3.1 is unavoidable in the task of learning cluster assignments, and by extension any instantiation of the cluster-then-predict approach.

**Beyond 2-component isotropic mixtures.** For discriminant calibration error, extending these results beyond 2 components to general $k$-component mixtures complicates the analysis in two ways. First, the ground-truth cluster assignment functions $x \mapsto \arg\max_{i\in[k]} f(i \mid x)$ are no longer halfspaces but rather Voronoi diagrams for $k > 2$. Second, the minimax optimal accuracy rate for estimating the cluster assignment function of a $k$-component isotropic Gaussian mixture model is not known exactly, aside from being polynomial (Belkin & Sinha, 2015). Extending these results to mixtures of non-isotropic Gaussians or to non-uniform mixtures is also non-trivial, even for $k = 2$; minimax optimal rates are similarly unknown for these generalizations. One challenge for proving such a result is that the cluster assignment functions are no longer halfspaces, but rather non-linear boundaries, as illustrated in Figure 1.

**Extension to likelihood calibration error.** Applying the cluster-then-predict approach to minimize likelihood calibration error again provides a $T^{2/3}$ rate.

**Proposition 3.4.** *Let $f$ be an unknown endogenous subgroups model whose Gaussian components are isotropic, and where $\mu_1$ and $\mu_2$ are separated by a constant in every dimension. Then, with probability $1 - \delta$, a* Cluster-Then-Predict Algorithm for Minimizing LCE*, setting $T' = O(T^{2/3})$, incurs likelihood calibration error of $\widetilde{O}\left(T^{2/3}\sqrt{d\log(d/\delta)}\right).$*

We defer the proof of Proposition 3.4, and the statement of the corresponding algorithm, to Appendix B.

Implementing this approach requires some additional care.

In the first phase, we must learn a good likelihood function for each cluster (i.e., $f(g \mid x)$), rather than a cluster assignment function. This can be done by estimating the parameters of each component, then using those estimates to construct likelihood functions; to that end, we can apply existing parameter learning algorithms (e.g. Hardt & Price (2015)). A more significant challenge is that even if a good estimator $\widehat{f}(g \mid x)$ is known, the fact that group membership is real-valued means that we can no longer partition the space and independently calibrate predictions in each partition. Instead, our predictions must handle the fact that each $x_t$ belongs to multiple groups; this motivates the use of online calibration algorithms with multi-group guarantees, called *multicalibration*. For similar reasons, we apply multicalibration algorithms in our multi-objective approach, which we discuss in the following section.

## 4. Improved bounds: A multi-objective approach

The cluster-then-predict approach studied in Section 3 necessitates learning the exact cluster structure underlying the data distribution—that is, learning the binary functions $x \mapsto \arg\max_g f(g \mid x)$ or the conditional likelihoods $f(g \mid x)$ to high accuracy. As an alternative to resolving the structure explicitly, we consider a multi-objective approach where we aim to simultaneously provide subgroup guarantees for a representative uncertainty set (specifically, a covering) of all possible subpopulation structures.

Building a covering of possible underlying cluster structures is significantly easier in a statistical sense than learning the true structure directly, which offers three benefits. First, rather than paying the $T^{2/3}$ error rate typical of cluster-then-predict methods, the multi-objective approach provides an optimal $T^{1/2}$ error rate. Second, the multi-objective learning approach can obtain a faster convergence rate than is possible when learning the underlying clusters. For example, rather than paying the inevitable mean separation dependence involved in learning discriminant or likelihood functions, the multi-objective approach provides error rates independent of separation. Third, while both the cluster-then-predict and multi-objective approaches can generalize beyond Gaussian mixture models, the latter does so without requiring distribution-specific clustering algorithms, which would have been necessary for the former; instead, the multi-objective approach only needs to consider the combinatorial structure of the likelihood functions. We provide tools for analyzing this structure in general (Lemma 4.6), and for exponential families in particular (Lemma 4.7).

In Section 4.1, we introduce multicalibration as a technical tool for our problem setting; in Section 4.2, we give an algorithm that provides guarantees for discriminant calibration error and likelihood calibration error by constructing a

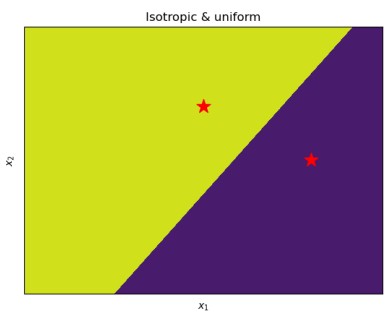 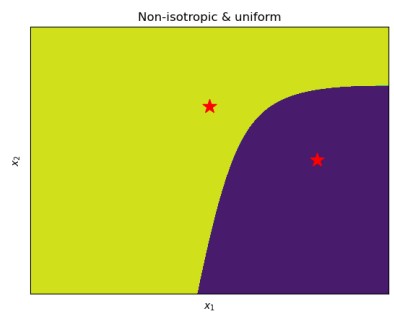 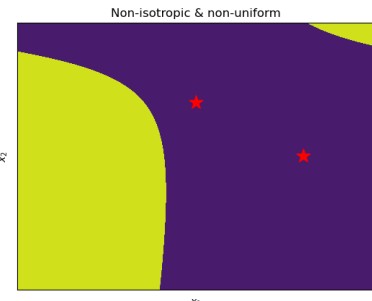

Figure 1: Illustration of the non-linear boundaries that arise in the cluster assignment functions of non-isotropic Gaussian mixtures for $k = 2$. Means of each component are marked with red stars.

cover of all plausible subpopulation structures. In Section 4.3, we show that, for the endogenous subgroups model, the key quantity to bound is the pseudodimension of likelihood ratios of the densities in $\mathcal{F}$, and that for exponential families, it in fact suffices to bound the dimension of the sufficient statistic. Finally, in Section 4.4, we provide concrete bounds on discriminant calibration error and likelihood calibration error in mixtures of exponential families. These results are summarized in Theorem 4.1.

**Theorem 4.1.** *Consider an endogenous subgroups model with $k$ exponential family mixture components and a sufficient statistic dimension of $\dim(T(x)) = d$. With probability $1 - \delta$, Algorithm 2, run with $\mathcal{G}$ as defined in Eq. 1, attains discriminant calibration error of*

$$O\big(\sqrt{T(dk \log(T) \log(k) + \log(\lambda/\delta))}\big).$$

*If Algorithm 2 is instead run with $\mathcal{G}$ as in Eq. 2, with probability $1 - \delta$, it attains likelihood calibration error of*

$$O\big(\sqrt{T(dk \log(T) \log^2(dk) + \log(\lambda/\delta))}\big).$$

### 4.1. Multicalibration as a tool for bounding discriminant and likelihood calibration error

Multicalibration is a refinement of calibration that requires unbiasedness of predictions not only on distinguishers resolving the level sets of one's predictors, i.e. $\{\mathbb{1}[p_t(x_t) \in v]\}_{v \in V_\lambda}$, but also on distinguishers that identify parts of the domain. We use an adaptation of the original multicalibration definition for real-valued distinguishers of the domain.

**Definition 4.2** (Multicalibration Error). **Given a sequence of instance-outcome pairs $(x_1, y_1), \ldots, (x_T, y_T)$ and class of distinguishers $\mathcal{G} \subset [0, 1]^{\mathcal{X}}$, the *multicalibration error* of predicted probabilities $\widehat{y}_1, \ldots, \widehat{y}_T$ with respect to $\mathcal{G}$ is**

$$\mathbf{MCE}(\widehat{y}_{1:T}, x_{1:T}, y_{1:T}; \mathcal{G}) :=$$

$$\max_{g \in \mathcal{G}} \max_{v \in V_\lambda} \left| \sum_{t \in [T]} g(x_t) \cdot \mathbb{1}[\widehat{y}_t \in v] \cdot (\widehat{y}_t - y_t) \right|.$$

When predictors $p_1, \ldots, p_T$ are used for making predictions $\widehat{y}_t = p_t(x_t)$, we denote the corresponding multicalibration error by $\mathbf{MCE}(p_{1:T}, x_{1:T}, y_{1:T}; \mathcal{G})$.

To show how multicalibration can be useful, note that we can upper bound likelihood calibration error and discriminant calibration error in terms of multicalibration error for carefully-designed classes of distinguishers.

*Fact* 4.3. Let $\mathcal{F}$ be the class of densities considered in an instantiation of the endogenous subgroups model. Take $f \in \mathcal{F}$ to be one such density and let $H := p_{1:T}, x_{1:T}, y_{1:T}$ be any transcript of predictors, instances, and labels. Then, $\mathbf{DCE}_f(H) \leq \mathbf{MCE}(H; \mathcal{G})$ defined for any set of distinguishers $\mathcal{G}$ where $\mathcal{G} \supseteq \{x \mapsto \mathbb{1}[g = \arg\max_j f(j \mid x)] \mid g \in [k]\}$. One such choice of $\mathcal{G}$ is

$$\mathcal{G} = \{x \mapsto \mathbb{1}[g = \arg\max_j \widetilde{f}(j \mid x)] \mid g \in [k], \widetilde{f} \in \mathcal{F}\}. \tag{1}$$

Similarly, we have $\mathbf{LCE}_f(H) \leq \mathbf{MCE}(H; \mathcal{G})$ for any set of distinguishers $\mathcal{G}$ where $\mathcal{G} \supseteq \{x \mapsto f(g \mid x) \mid g \in [k]\}$. One such choice of $\mathcal{G}$ is

$$\mathcal{G} = \{x \mapsto \widetilde{f}(g \mid x) \mid g \in [k], \widetilde{f} \in \mathcal{F}\}. \tag{2}$$

### 4.2. Multicalibration algorithms for distinguisher classes of finite pseudodimension

In this section, we use multicalibration algorithms to simultaneously provide per-subgroup guarantees for multiple hypothetical clustering schemes, a multi-objective approach that does not require resolving the underlying clustering.

For any (potentially infinite) set of real-valued distinguishers $\mathcal{G}$ with finite pseudodimension, the following algorithm achieves $O(\sqrt{T})$ multicalibration error (Theorem 4.4): efficiently cover the space of distinguishers and run a standard online multicalibration algorithm on the cover. In Appendix C, we give explicit example algorithms for each stage—the covering stage (Algorithm 3) and the calibration stage (Algorithm 4).

Algorithm 2: *Online Multicalibration Algorithm for Coverable Distinguishers*

For the first $T' = \sqrt{T(\mathrm{Pdim}(\mathcal{G})\log(T) + \log(1/\delta))}$ timesteps, make arbitrary predictions and collect observed features $x_1, \ldots, x_{T'}$ and compute a small $\frac{1}{T'}$-covering $\mathcal{G}'$ of the set $\mathcal{G}$ on $x_1, \ldots, x_{T'}$, e.g. using Algorithm 3.

For the remaining timesteps $t = T' + 1, \ldots, T$, observe $x_t$ and predict $y_t$ by applying any online multicalibration algorithm, e.g. Algorithm 4, to the transcript of previously seen datapoints $\{(x_\tau, y_\tau) \mid \tau < t\}$ and distinguishers $\mathcal{G}'$.

Theorem 4.4 bounds the error incurred by Algorithm 2.

**Theorem 4.4.** *For any real-valued function class $\mathcal{G}$ with finite pseudodimension $\mathrm{Pdim}(\mathcal{G})$, with probability $1 - \delta$, the predictions $p_{1:T}$ made by Algorithm 2 satisfy* $\mathbf{MCE}(p_{1:T}, x_{1:T}, y_{1:T}; \mathcal{G}) \leq \widetilde{O}\left(\sqrt{T(\mathrm{Pdim}(\mathcal{G}) + \log(\lambda/\delta))}\right).$

The proof of Theorem 4.4 requires two main ingredients. First, standard analysis of multicalibration algorithms (e.g. Haghtalab et al. (2023) Theorem 4.3—see Theorem C.1) suggests that a $\sqrt{T}$ rate is possible when the algorithm is run with a *finite* set of distinguishers. To obtain such a set, Algorithm 2 first spends $T'$ steps to compute a $\frac{1}{T}$-covering of the (infinite) $\mathcal{G}$. The second ingredient, therefore, is to show that such a finite covering $\mathcal{G}'$ of the (infinite) $\mathcal{G}$ can be computed using a small number of observations, so that the error incurred by this approximation in steps $T' + 1 \ldots T$ is sufficiently small. To that end, we give the following lemma, which states that an empirical cover computed on a finite number of samples is a true $\varepsilon$-cover with high probability.

**Lemma 4.5.** *Given $\varepsilon, \delta, T > 0$ and a real-valued function class $\mathcal{G}$ with finite pseudodimension $\mathrm{Pdim}(\mathcal{G})$, consider any $\varepsilon$-cover $\mathcal{G}'$ of $\mathcal{G}$ computed on $T$ randomly sampled datapoints. Then $\mathcal{G}'$ is a $4\varepsilon$-cover of $\mathcal{G}$ with probability at least $1 - O\left(\mathrm{Pdim}(\mathcal{G})^2(1/\varepsilon)^{O(\mathrm{Pdim}(\mathcal{G}))}\exp(-T\varepsilon)\right).$*

We prove Theorem 4.4 and Lemma 4.5 in Appendices C.2 and C.3, respectively.

### 4.3. Bounding the pseudodimension of DCE and LCE distinguishers

We now turn to instantiating our multi-objective approach for minimizing both discriminant calibration error and likelihood calibration error. Fact 4.3 and Theorem 4.4 suggest that it suffices to construct a class of distinguishers $\mathcal{G}$ that includes the relevant likelihood or discriminant functions,

then bound its pseudodimension. In this section, we show that for an endogenous subgroups model instantiated with density class $\mathcal{F}$, a bound on the pseudodimension of the set of possible likelihood ratios,

$$\mathcal{H} = \left\{ x \mapsto \frac{f(1 \mid x)}{f(j \mid x)} \mid j \in [k], f \in \mathcal{F} \right\}, \qquad (3)$$

is the key ingredient for bounding $\mathrm{Pdim}(\mathcal{G})$ for both discriminant calibration error (i.e., with $\mathcal{G}$ as defined in (1)) and likelihood calibration error (i.e., with $\mathcal{G}$ as defined in (2)). We formalize this in Lemma 4.6.

**Lemma 4.6.** *Let $D := \mathrm{Pdim}(\mathcal{H})$ where $\mathcal{H}$ is defined as in (3). Then, the binary-valued distinguisher class $\mathcal{G} = \{x \mapsto \mathbb{1}[g = \arg\max_{g' \in [k]} \widetilde{f}(g' \mid x)] \mid g \in [k], \widetilde{f} \in \mathcal{F}\}$ has pseudodimension (equivalently VC dimension) of at most $O(kD\log(k))$, and the real-valued distinguisher class $\mathcal{G}' = \{x \mapsto \widetilde{f}(g \mid x) \mid g \in [k], \widetilde{f} \in \mathcal{F}\}$ has pseudodimension at most $O(kD\log^2(kD))$.*

Note that Lemma 4.6 holds for any class of densities $\mathcal{F}$. In the following section, we show how to bound $\mathrm{Pdim}(\mathcal{H})$ for exponential families and Gaussian mixture models.

### 4.4. Concrete bounds for endogenous subgroups models with exponential families

We now prove Theorem 4.1, and discuss some of its implications. As suggested in Section 4.3, a key step in the proofs of Theorem 4.1 is to bound the pseudodimension of $\mathcal{H}$. When $\mathcal{F}$ is an exponential family, we can bound the pseudodimension of likelihood ratios, and by extension $\mathrm{Pdim}(\mathcal{H})$, by bounding the dimension of its sufficient statistic $T(x)$.

**Lemma 4.7.** *When $\mathcal{F}$ is an exponential family (Definition 2.4), the pseudodimension of $\mathcal{H}$ (with $\mathcal{H}$ defined as in (3)) is bounded by the dimension of the sufficient statistic, i.e. $\mathrm{Pdim}(\mathcal{H}) \leq \dim(T(x)) + 1.$*

*Proof of Theorem 4.1.* Lemma 4.7 upper bounds the pseudodimension of $\mathcal{H}$ (3) for exponential families, and thus also provides a bound on the pseudodimension of distinguisher classes $\mathcal{G}$ and $\mathcal{G}'$ in Lemma 4.6. Theorem 4.1 then follows by Theorem 4.4's upper bound on multicalibration error, and therefore discriminant calibration error and likelihood calibration error (Fact 4.3). $\square$

As a consequence of Theorem 4.1, we can explicitly bound discriminant calibration error and likelihood calibration error when the underlying mixture components are Gaussian, in light of the following observation about the dimension of the sufficient statistic for Gaussian densities.

*Fact* 4.8. For Gaussian densities, $\dim(T(x)) \leq \frac{d(d+1)}{2} + d$. In the isotropic case, $\dim(T(x)) \leq 2d$.

Fact 4.8 and Theorem 4.1 immediately imply the following bounds on discriminant calibration error and likelihood calibration error for Gaussian mixtures.

**Corollary 4.9.** *Consider an endogenous subgroups model with $k$ Gaussian mixture components. With probability $1 - \delta$, Algorithm 2, run with $\mathcal{G}$ as defined in Eq. 1, attains discriminant calibration error of $O\big(\sqrt{T(d^2 k \log(T) \log(k) + \log(\lambda/\delta))}\big)$ or, under isotropic assumptions, $O\big(\sqrt{T(dk \log(T) \log(k) + \log(\lambda/\delta))}\big)$.*

*On the other hand, when run with $\mathcal{G}$ as defined in Eq. 2, Algorithm 2 instead attains likelihood calibration error of $O\big(\sqrt{T(d^2 k \log(T) \log^2(dk) + \log(\lambda/\delta))}\big)$ or, under isotropic assumptions, $O\big(\sqrt{T(dk \log(T) \log^2(dk) + \log(\lambda/\delta))}\big)$.*

**A gap between learning subgroups and providing per-subgroup guarantees.** In contrast to the Cluster-then-Predict guarantees of Section 3 (Propositions 3.1 and 3.4), Corollary 4.9 achieves an improved error rate of $\widetilde{O}(\sqrt{T})$ versus $\widetilde{O}(T^{2/3})$, while also avoiding separation dependence—for both discriminant calibration error and likelihood calibration error.

For discriminant calibration error, Corollary 4.9 is especially notable because *learning* the subgroup discriminator function requires sample complexity that scales with component mean separation.

**Theorem 4.10** (Azizyan et al. (2013), Theorem 2)**.** *Let $f$ be an unknown endogenous subgroups model with two isotropic Gaussian components with $d \geq 9$. The sample complexity of learning the cluster assignment function is $\Omega\big(\frac{d}{\varepsilon^2 \gamma^6}\big)$.*

Notably, this lower bound holds even for the case of having two equal-weight isotropic Gaussians. This sample complexity is paid for, for example, in the first stage of the Cluster-then-Predict approach.

Similarly, the likelihood calibration error rate in Corollary 4.9 is also significantly better than the best known rates for learning multivariate Gaussian likelihood functions: Hardt & Price (2015) prove an $\varepsilon^{-12}$ lower bound for learning parameters in a mixture of two Gaussians, which would suggest a corresponding $T^{12/13}$ rate; a $T^{1/2}$ rate would have only been achievable under the separation assumption of Proposition 3.4.

That Corollary 4.9 obtains a better bound highlights the surprising fact that *providing clusterable group guarantees can be easier than clustering*, lending further motivation to multi-objective approaches over cluster-then-predict.

## 5. Discussion

This work has focused on a particular instantiation of our model—for (online) calibration as the objective, and mixtures of exponential families as the subgroup structure underlying our endogenous subgroups model. However, as discussed in Section 2, the results presented in this paper for calibration extend to other problems that can be formalized in the language of Blackwell approachability, such as online conformal prediction. Another extension is to handle other subgroup structures beyond exponential families. While Lemma 4.7 is specific to exponential families, Lemma 4.6 is more general; in principle, we expect that similar technical analysis can be performed for other unsupervised learning models of bounded combinatorial complexity. In fact, our approaches—and notions of discriminant calibration error and likelihood calibration error—can apply to any setting where group membership can at best be estimated.

More generally, our formalization of an unsupervised notion of multi-group guarantees provides a language for understanding an important downstream application of clustering. Our results demonstrate that being intentional about how learned clusters will be used, rather than treating clustering and learning as distinct stages, is significant both conceptually and for attaining optimal theoretical rates. First, resolving the exact clustering structure of one's data is inefficient, and results in the same theoretical sub-optimality as explore-then-commit algorithms in bandit and RL literature—namely, $O(T^{2/3})$ rather than $O(T^{1/2})$ rates. Second, the task of learning with guarantees for subgroups can be surprisingly easier than learning the subgroups themselves. The most striking example of this appears in our results for discriminant calibration error, for which we show that learning cluster assignment functions has an inevitable dependence on cluster separation, whereas separation can be ignored when pursuing per-cluster guarantees. Moreover, as our improved rates for the multi-objective approach suggest, it is not just that learning subgroups may be harder: it is also not necessary to learn the subgroups exactly if the ultimate goal is to provide guarantees across them.

## Acknowledgments

This work was supported in part by the National Science Foundation under grant CCF-2145898, by the Office of Naval Research under grant N00014-24-1-2159, a C3.AI Digital Transformation Institute grant, and Alfred P. Sloan fellowship, and a Schmidt Science AI2050 fellowship. This material is based upon work supported by the National Science Foundation Graduate Research Fellowship Program under Grant No. DGE 2146752 (EZ and JD). Any opinions, findings, and conclusions or recommendations expressed in this material are those of the author(s) and do not necessarily reflect the views of the National Science Foundation.

## Impact Statement

This work seeks to provide a new perspective on how subpopulations should be understood. We do not feel that the potential societal impacts of this work merit additional commentary beyond what is already discussed in the paper and appendices.

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

# A. Facts, references, and restatements

*Fact* A.1 (Anthony & Bartlett (1999), Theorem 18.4). Let $\mathcal{G}$ be a real-valued function class. Then $N_1(\varepsilon, \mathcal{G}, m) \leq O\left(\text{Pdim}(\mathcal{G})(1/\varepsilon)^{\text{Pdim}(\mathcal{G})}\right)$. If $\mathcal{G}$ is a binary function, the bound holds for $\text{VC}(\mathcal{G})$.

*Fact* A.2 (Anthony & Bartlett (1999), Theorem 11.4). The pseudodimension of a $k$-dimensional vector space of real valued functions is $k + 1$.

*Fact* A.3 (Anthony & Bartlett (1999), Theorem 11.3). Let $f$ be a monotonic function and $\mathcal{G}$ be a real-valued function class with pseudodimension $d$. Then the pseudodimension of $\{f \circ g \mid g \in \mathcal{G}\}$ is $d$.

*Fact* A.4 (Attias & Kontorovich (2024), Theorem 1). Let $\mathcal{G}_1, \ldots, \mathcal{G}_k$ be real-valued function classes each with a pseudodimension of $d$. Then the pseudodimension of the classes $\{\sum_i f_i \mid f_1 \in \mathcal{G}_1, \ldots, f_k \in \mathcal{G}_k\}$ and $\{\max_i f_i \mid f_1 \in \mathcal{G}_1, \ldots, f_k \in \mathcal{G}_k\}$ is $O(kd \log^2(kd))$.

*Fact* A.5 (Devroye et al. (2013), Theorem 21.5). The VC dimension of the class of $k$-cell Voronoi diagrams in $\mathbb{R}^d$ is upper bounded by $k + (d + 1)k^2 \log k$.

*Fact* A.6 (Blumer et al. (1989), Lemma 3.2.3). The intersection of $k$ concept classes with VC dimension $D$ has VC dimension at most $O(kD \log(k))$.

# B. Proofs for Section 3

## B.1. Proposition 3.1

Let $\Psi_\gamma = \{\mu_1, \mu_2 \in \mathcal{X} \mid \|\mu_1 - \mu_2\| \geq \gamma\}$ denote the space of possible component means with at least $\gamma$ separation. Let $\mathcal{F}_n$ be the class of all mixture model estimators; formally, we define $\mathcal{F}_n$ as the set of all functions mapping from $n$-length datasets $(\mathcal{X})^n$ to functions $\{1, 2\}^{\mathcal{X}}$. Azizyan et al. (2013) provides an estimator for the Gaussian mixture model that achieves the minimax optimal error rate, with a guarantee as follows.

**Theorem B.1** (Minimax Gaussian clustering rates (Azizyan et al., 2013)). *For $n \geq \max\{68, 4d\}$, the minimax optimal accuracy for the estimator of a two-component isotropic Gaussian mixture model with separation $\gamma$ is*

$$\inf_{F \in \mathcal{F}_n} \sup_{\theta \in \Psi_\gamma} \mathop{\mathbb{E}}_{\{x_1, \ldots, x_n\} \sim \mathcal{D}_\theta^n} \left[ \Pr_{x \sim \mathcal{D}_\theta} \left[ F(x_1, \ldots, x_n)(x) \neq \arg\max_{i \in \{1,2\}} f(g = i \mid x) \right] \right] \in \widetilde{\Theta}\left( \frac{1}{\gamma^2} \sqrt{\frac{d}{n}} \right)$$

*where $\widetilde{\Theta}$ suppresses logarithmic factors, $\mathcal{D}_{\mu_1, \mu_2}$ denotes the uniform mixture of $\mathcal{N}(\mu_1, I_d)$ and $\mathcal{N}(\mu_2, I_d)$, and $\mathcal{D}_{\mu_1, \mu_2}^n$ denotes $n$ i.i.d. samples from $\mathcal{D}_{\mu_1, \mu_2}$.*

*Proof of Proposition 3.1.* We instantiate the cluster-then-predict algorithm with the Gaussian mixture model estimator of (Azizyan et al., 2013) for the first phase. For the second phase, we use the multicalibration algorithm of Algorithm 4; we will instantiate Algorithm 4 with the trivial distinguisher set $\mathcal{G}_1 = \{x \to 1\}$ because we only need calibration with marginal guarantees within each bucket. By Theorem B.1, the expected clustering error attained by the (Azizyan et al., 2013) estimator learned with $T'$ samples is $O\left(\frac{1}{\gamma^2} \sqrt{\frac{d}{T'}}\right)$. Let the resulting cluster assignment function be denoted $F$. Using Hoeffding's inequality, this implies that with probability at least $1 - \delta$,

$$\sum_{t=T'}^{T} \mathbb{1}\left[ F(x_t) \neq \arg\max_{j \in [k]} f(j \mid x_t, y_t) \right] \leq \mathbb{E}\left[ \sum_{t=T'}^{T} \mathbb{1}\left[ F(x_t) \neq \arg\max_{j \in [k]} f(j \mid x_t, y_t) \right] \right] + \sqrt{(T - T') \log(1/\delta)}$$

$$\leq O\left( \frac{1}{\gamma^2} \sqrt{\frac{d}{T'}}(T - T') + \sqrt{(T - T') \log(1/\delta)} \right). \tag{4}$$

With a slight abuse of notation, let us define the quantity $\mathbf{DCE}_F(\widehat{y}_{T':T}, x_{T':T}, y_{T':T})$ as the discriminant calibration error that *would have* been incurred had $F : \mathcal{X} \to \{1, 2\}$ been the true discriminant with respect to $f$, that is,[2]

$$\mathbf{DCE}_F(\widehat{y}_{T':T}, x_{T':T}, y_{T':T}) := \max_{g \in [k]} \max_{v \in V_\lambda} \left| \sum_{t=1}^{T} \mathbb{1}\left[ g = F(x_t) \right] \cdot \mathbb{1}\left[ \widehat{y}_t \in v \right] \cdot (\widehat{y}_t - y_t) \right|.$$

---

[2]Note that the in the usual definition of $\mathbf{DCE}$, the only information needed about the endogenous subgroups model $f$ is the corresponding discriminant function $\arg\max_j f(j|x, y)$, and $f(j|x, y)$ is independent of $y$ given $x$.

By Theorem C.1, the calibration error on timesteps $t > T'$ where cluster 1 is predicted, i.e. $F(x_t) = 1$, is bounded with probability at least $1 - \delta$ by $O(\sqrt{T_1 \log(1/\delta)})$ where $T_1 = \sum_{\tau=T'+1}^{T} \mathbb{1}[F(x_t) = 1]$ is the number of timesteps where cluster 1 is predicted. With a similar bound holding for cluster 2, by union bound, we have that

$$\mathbf{DCE}_F(\widehat{y}_{T':T}, x_{T':T}, y_{T':T}) \in O(\sqrt{(T - T') \log(1/\delta)}). \tag{5}$$

By triangle inequality and an additional union bound, combining (4) and (5) gives

$$\mathbf{DCE}_f(\widehat{y}_{1:T}, x_{1:T}, y_{1:T}) \leq O\left(T' + \tfrac{1}{\gamma^2}\sqrt{\tfrac{d}{T'}}(T - T') + \sqrt{(T - T') \log(1/\delta)}\right).$$

Choosing $T' = \Theta(d^{1/3}T^{2/3}\gamma^{-4/3})$ gives the desired upper bound of

$$\mathbf{DCE}_f(\widehat{y}_{1:T}, x_{1:T}, y_{1:T}) \leq O\left(d^{1/3}T^{2/3}\gamma^{-4/3} + \sqrt{T \log(1/\delta)}\right).$$

$\square$

## B.2. Proposition 3.4

In this section, we prove a generalization of Proposition 3.4: Theorem B.2.

---

### *Cluster-Then-Predict Algorithm for Minimizing* **LCE**

For the first $T' < T$ timesteps, make arbitrary predictions and collect observed features $x_1, \ldots, x_{T'}$. Apply a parameter-learning algorithm, such as the (Hardt & Price, 2015) method, to the observed features to obtain estimates of the per-component likelihoods $\widehat{f}(x \mid g)$ for each $g \in [k]$.

Then, instantiate Algorithm 4 with distinguishers $\mathcal{G} = \left\{x \mapsto \widehat{f}(g \mid x) \mid g \in [k]\right\}$. For each timestep $t = T' + 1, \ldots, T$, observe $x_t$ and predict $y_t$ by applying Algorithm 4 to the transcript of previously seen datapoints $\{(x_\tau, y_\tau) \mid T' < \tau < t\}$.

---

While the algorithm is written for general $k$ and $f$, we focus on the case where $k = 2$ and $w_1 = w_2 = 1/2$.

**Theorem B.2.** *Let $k = 2$ and $w_1 = w_2 = 1/2$. Define $\sigma = \|\mu_1 - \mu_2\|_\infty^2 + \|\Sigma_1\|_\infty + \|\Sigma_2\|_\infty$. If we have that $\min_{j \in [d]} |\mu_{1,j} - \mu_{2,j}| \geq \Omega(\sigma)$, then, with probability $1 - \delta$, the* Cluster-Then-Predict Algorithm for Minimizing **LCE**, *setting $T' = O(T^{2/3})$, incurs*

$$\mathbf{LCE}_f(p_{1:T}, x_{1:T}, y_{1:T}) \leq \widetilde{O}\left(T^{2/3}\sqrt{d}\log^{1/2}(d/\delta)\right).$$

*On the other hand, without separation, we must set $T' = O(T^{12/13})$ and incur*

$$\mathbf{LCE}_f(p_{1:T}, x_{1:T}, y_{1:T}) \leq \widetilde{O}(T^{12/13}\sqrt{d}\log^{1/2}(d/\delta)).$$

To prove Theorem B.2, we will need some facts relating parameter estimation to errors in estimating group membership are bounded by $O(\varepsilon)$.

*Fact* B.3. When parameters $(\mu_i, \Sigma_i)$ of mixture component $i$ are $\varepsilon, \delta$-learned in the sense of (Hardt & Price, 2015), we have that with probability $1 - \delta$, $\mathrm{TV}\left(f(x \mid g_i), \widehat{f}(x \mid g_i)\right) \leq O(\varepsilon\sqrt{d})$.

*Proof of Fact B.3.* To see this, note that $\varepsilon, \delta$-learning in (Hardt & Price, 2015) is defined in terms of the $\ell_\infty$-norm on the estimated parameters relative to the variance of the mixture, i.e.

$$\max_{i=\{1,2\}} \max\left(\|\mu_i - \widehat{\mu}_i\|_\infty^2, \|\Sigma_i - \widehat{\Sigma}_i\|_\infty\right) \leq \varepsilon^2\left(\|\mu_1 - \mu_2\|_\infty^2 + \|\Sigma_1\|_\infty + \|\Sigma_2\|_\infty\right).$$

Theorem 1.8 of (Arbas et al., 2023) shows that $\mathrm{TV}\left(\mathcal{N}(\mu, \Sigma), \mathcal{N}(\widehat{\mu}, \widehat{\Sigma})\right) = \Theta(\Delta)$ where $\Delta$ is parameter distance in $\ell_2$; specifically,

$$\Delta = \max\left(\|\Sigma^{-1/2}\widehat{\Sigma}\Sigma^{-1/2} - I_d\|_F, \|\Sigma^{-1/2}(\mu - \widehat{\mu})\|_2\right).$$

Translating between the $\ell_2$ and $\ell_\infty$ norms incurs a $O(\sqrt{d})$ penalty. $\qquad\square$

*Fact* B.4. Suppose that for each component $g_i$, we have estimates of the density of $x$ conditioned on membership in $g_i$, i.e. $\widehat{f}(x \mid g_i)$, such that $\mathrm{TV}\left(f(x \mid g_i), \widehat{f}(x \mid g_i)\right) \le \varepsilon$. Then, mistakes in estimating group membership can be bounded as $\mathbb{E}_{x \sim f}\left[\left|f(g_i \mid x) - \widehat{f}(g_i \mid x)\right|\right] \le 3\varepsilon$, and the overall TV distance between the true mixture and the estimated mixture can be bounded as $\mathrm{TV}(f(x), \widehat{f}(x)) \le \varepsilon$.

*Proof of Fact B.4.* By the definition of conditional probability, we have $f(g_i \mid x) = \frac{f(x,g_i)}{f(x)}$ for every $g_i$. Then, we can write

$$\mathbb{E}\left[\left|f(g_i \mid x) - \widehat{f}(g_i \mid x)\right|\right] = \int \left|f(g_i \mid x) - \widehat{f}(g_i \mid x)\right| f(x)dx$$

$$= \int \left|f(x,g_i) - \widehat{f}(x,g_i) - \left(\frac{f(x)}{\widehat{f}(x)} - 1\right)\widehat{f}(x,g_i)\right| dx$$

$$\le \underbrace{\int \left|f(x,g_i) - \widehat{f}(x,g_i)\right| dx}_{(A)} + \underbrace{\int \left|\left(\frac{f(x)}{\widehat{f}(x)} - 1\right)\widehat{f}(x,g_i)\right| dx}_{(B)}.$$

Again by the definition of conditional probability, $f(x,g_i) = f(x \mid g_i) \cdot f(g_i)$, and likewise for the estimated quantity. Recall that the marginal likelihood of $g_i$ (i.e. its mixing weight) is $f(g_i) = \frac{1}{2}$. Then, $(A)$ reduces to $2 \cdot \frac{1}{2} \cdot \mathrm{TV}\left(f(x \mid g_i), \widehat{f}(x \mid g_i)\right) \le \varepsilon$. For $(B)$, noting that $\widehat{f}(g_i \mid x) \le 1$ for all $x$, we have

$$\int \left|\left(\frac{f(x)}{\widehat{f}(x)} - 1\right)\widehat{f}(x,g_i)\right| dx = \int \left|f(x) - \widehat{f}(x)\right| \frac{\widehat{f}(x,g_i)}{\widehat{f}(x)}dx$$

$$= \int \left|f(x) - \widehat{f}(x)\right| \widehat{f}(g_i \mid x)dx$$

$$\le \int \left|f(x) - \widehat{f}(x)\right| dx.$$

Recalling that $f(x) = \sum_{i \in [k]} f(x \mid g_i) \cdot f(g_i)$ and $k = 2$, we can bound the final quantity in the above display by $2\varepsilon$. $\qquad\square$

A direct consequence of Fact B.4 is that the **LCE** incurred by a sequence of predictors $p_1, \ldots, p_T$ on samples $(x, y)$ from the true distribution is close to the **LCE** that would have been incurred had each $(x, y)$ been sampled from the estimated distribution.

**Lemma B.5.** *Suppose that for each component $g_i$, we have estimates of the density of $x$ conditioned on membership in $g_i$, i.e. $\widehat{f}(x \mid g_i)$, such that $\mathrm{TV}\left(f(x \mid g_i), \widehat{f}(x \mid g_i)\right) \le \varepsilon$. Then, with probability $1 - \delta$, the **LCE** incurred by a fixed sequence of predictors $p_1, \ldots, p_T$ on datapoints $(x, y) \sim f$ can be bounded as*

$$\mathbf{LCE}_f(p_{1:T}, x_{1:T}, y_{1:T}) \le \mathbb{E}_{\widehat{f}}[\mathbf{LCE}_{\widehat{f}}(p_{1:T}, x_{1:T}, y_{1:T})] + O(\sqrt{T})\ln(1/\delta) + 5T\varepsilon.$$

*Proof.* First, by definition, we have

$$\mathbf{LCE}(p_1, \ldots, p_T) = \max_{g \in [k]} \max_{v \in V_\lambda} \left|\sum_{t \in [T]} f(g \mid x_t) \cdot \mathbb{1}[p_t(x_t) \in v] \cdot (p_t(x_t) - y_t)\right|.$$

Note that at each timestep $t$, and for every $g$ and $v$, the quantity $f(g \mid x_t) \cdot \math1[p_t(x_t) \in v] \cdot (p_t(x_t) - y_t)$ is a random variable bounded in $[0, 1]$. Therefore, for any $g$ and $v$, we can relate the realized sum to its expected sum; in particular, with probability $1 - \delta$,

$$\left| \sum_{t \in [T]} f(g \mid x_t) \cdot \math1[\widehat{y}_t \in v] \cdot (\widehat{y}_t - y_t) \right| \leq O(\sqrt{T}) \ln(1/\delta) + \left| \mathbb{E}\left[ \sum_{t \in [T]} f(g \mid x_t) \cdot \math1[p_t(x_t) \in v] \cdot (p_t(x_t) - y_t) \right] \right|$$

$$= O(\sqrt{T}) \ln(1/\delta) + \left| \sum_{t \in [T]} \int f(g \mid x) \cdot \math1[p_t(x) \in v] \cdot (p_t(x) - y) f(x) dx \right|$$

$$\leq O(\sqrt{T}) \ln(1/\delta)$$
$$+ T \int \left| f(g_i \mid x) - \widehat{f}(g_i \mid x) \right| \cdot \math1[p_t(x) \in v] \cdot |p_t(x) - y| f(x) dx$$
$$+ \left| \sum_{t \in [T]} \int \widehat{f}(g_i \mid x) \cdot \math1[p_t(x) \in v] \cdot (p_t(x) - y) \cdot f(x) dx \right|, \tag{6}$$

where Eq. 6 is due to the triangle inequality. Now, we can express the first term in terms of the TV distance between the true and the estimated group-conditional densities. Combining Fact B.4 (for the first term of (6)) and the triangle inequality (for the second term of (6)), we have

$$(6) \leq O(\sqrt{T}) \ln(1/\delta) + 3T\varepsilon + \left| \sum_{t \in [T]} \int \widehat{f}(g \mid x) \cdot \math1[p_t(x) \in v] \cdot (p_t(x) - y) \cdot \widehat{f}(x) dx \right|$$

$$+ \sum_{t \in [T]} \int \widehat{f}(g \mid x) \cdot \math1[p_t(x) \in v] \cdot |p_t(x) - y| \cdot \left| f(x) - \widehat{f}(x) \right| dx$$

$$\leq O(\sqrt{T}) \ln(1/\delta) + 5T\varepsilon + \left| \sum_{t \in [T]} \int \widehat{f}(g \mid x) \cdot \math1[p_t(x) \in v] \cdot (p_t(x) - y) \cdot \widehat{f}(x) dx \right| \tag{7}$$

$$= O(\sqrt{T}) \ln(1/\delta) + 5T\varepsilon + \left| \mathbb{E}_{\widehat{f}}\left[ \sum_{t \in [T]} \widehat{f}(g \mid x) \cdot \math1[p_t(x) \in v] \cdot (p_t(x) - y) \right] \right|$$

$$\leq O(\sqrt{T}) \ln(1/\delta) + 5T\varepsilon + \mathbb{E}_{\widehat{f}}\left[ \left| \sum_{t \in [T]} \widehat{f}(g \mid x) \cdot \math1[p_t(x) \in v] \cdot (p_t(x) - y) \right| \right], \tag{8}$$

where Eq. (7) again comes from combining Fact B.4 with the triangle inequality and Eq. (8) is due to Jensen's inequality. The statement of the lemma follows by noting that, because this held for any $g$ and $v$, it also holds for the max $g$ and $v$; furthermore, by another application of Jensen's inequality,

$$\max_{g \in [k]} \max_{v \in V_\lambda} \mathbb{E}_{\widehat{f}}\left[ \left| \sum_{t \in [T]} \int \widehat{f}(g \mid x_t, y_t) \cdot \math1[\widehat{y}_t \in v] \cdot (\widehat{y}_t - y_t) \right| \right]$$

$$\leq \mathbb{E}_{\widehat{f}}\left[ \max_{g \in [k]} \max_{v \in V_\lambda} \left| \sum_{t \in [T]} \int \widehat{f}(g \mid x_t, y_t) \cdot \math1[\widehat{y}_t \in v] \cdot (\widehat{y}_t - y_t) \right| \right]$$

$$= \mathbb{E}_{\widehat{f}}[\mathbf{LCE}_{\widehat{f}}(p_{1:T}, x_{1:T}, y_{1:T})].$$

$\square$

We conclude this section with the proof of Theorem B.2.

*Proof of Theorem B.2.* The proof of Theorem B.2 proceeds in three steps. In Step 1, we analyze the estimation phase and show that spending $T'$ samples allows us to estimate $\widehat{f}(\cdot)$ with sufficiently small error. In Step 2, we analyze the calibration phase and relate the error incurred when using $\widehat{f}(\cdot)$ to the true error. Step 3 completes the argument.

**Step 1: Analyzing the estimation phase.** In the clustering phase of *Cluster-Then-Predict Algorithm for Minimizing* **LCE**, $T'$ samples are used to learn $\widehat{f}(x \mid g_i)$ for $i \in \{1, 2\}$. Let $\delta_1$ be the likelihood that parameters are successfully learned in this step.

Let $\sigma = \|\mu_1 - \mu_2\|_\infty^2 + \|\Sigma_1\|_\infty + \|\Sigma_2\|_\infty$. If we have that $\min_{j \in [d]} |\mu_{1,j} - \mu_{2,j}| \geq \Omega(\sigma)$, i.e., that $\mu_1$ and $\mu_2$ are sufficiently separated in all dimensions, then set $T' = \lceil T^{2/3} \rceil$ and the algorithm of (Hardt & Price, 2015) will $\varepsilon, \delta_1$-learn the parameters $(\mu_1, \Sigma_1)$ and $(\mu_2, \Sigma_2)$ with $\widetilde{O}(\varepsilon^{-2} \log(d/\delta))$ samples (where the $\widetilde{O}$ suppresses a $\log\log(1/\varepsilon)$ term). Setting $T^{2/3} = \widetilde{O}\left(\varepsilon^{-2} \log(d/\delta_1)\right)$, we have $\varepsilon = \widetilde{O}\left(T^{-1/3} \log^{1/2}(d/\delta_1)\right)$.

On the other hand, when $\mu_1$ and $\mu_2$ are not separated, we need $\widetilde{O}(\varepsilon^{-12} \log(d/\delta_1))$ samples to learn parameters. We set $T' = \lceil T^{12/13} \rceil$ and instead have $\varepsilon = \widetilde{O}(T^{-1/13} \log^{1/12}(d/\delta_1))$.

Finally, to analyze the error incurred in this phase, note that since the predictor $p_t(x_t) = \frac{1}{2}$ for each $t \leq T_1$,

$$\left| \sum_{t \in [T']} f(g_i \mid x_t) \cdot \mathbb{1}[p_t(x_t) \in v] \cdot (p_t(x_t) - y_t) \right| \leq \tfrac{1}{2} T^{2/3}$$

for any $i$ and $v$.

**Step 2: Analyzing error incurred in the calibration phase when using $\widehat{f}(\cdot)$.** In the calibration phase of *Cluster-Then-Predict Algorithm for Minimizing* **LCE**, the predictor $p_t$ is updated using the estimated densities $\widehat{f}(\cdot)$. Note that $\varepsilon$ error in parameter learning translates to $\varepsilon\sqrt{d}$ error in TV distance between the estimated $\widehat{f}(x \mid g_i)$ and the true $f(x \mid g_i)$, by Fact B.3. Therefore, in the event that all parameters are learned within additive error of $\varepsilon$ (which occurs with probability $1 - \delta_1$), Lemma B.5 combined with Fact B.3 gives us that, with probability $1 - \delta_2$, the error incurred from $t = T' + 1 \ldots T$ is at most

$$\left| \sum_{t = T', \ldots, T} f(g \mid x_t) \cdot \mathbb{1}[\widehat{y}_t \in v] \cdot (\widehat{y}_t - y_t) \right| \leq \mathbb{E}[\mathbf{LCE}_{\widehat{f}}(p_{T':T}, x_{T':T}, y_{T':T})] + 5T\varepsilon\sqrt{d} + O(\log(1/\delta_2)\sqrt{T - T'}).$$

Recall that we have defined $\mathcal{G} = \left\{ x \mapsto \widehat{f}(g \mid x) \mid g \in [k] \right\}$; note that $|\mathcal{G}| = k$. Then, by definition, $\mathbb{E}[\mathbf{LCE}_{\widehat{f}}(p_{T':T}, x_{T':T}, y_{T':T})] = \mathbb{E}[\mathbf{MCE}(p_{T':T}, x_{T':T}, y_{T':T}; \mathcal{G})]$.

We will now extend Theorem C.1 to hold for expected multicalibration error, i.e. $\mathbb{E}[\mathbf{MCE}(p_{T':T}, x_{T':T}, y_{T':T}; \mathcal{G})]$, rather than for specific realizations of $x_t, y_t$. In particular, integrating over $\delta \in (0, 1)$, we have that $\mathbb{E}[\mathbf{MCE}(p_{T':T}, x_{T':T}, y_{T':T}; \mathcal{G})] \leq O(\eta\sqrt{T - T' \log(k)})$ for $T - T' \geq C\eta^{-2} \log(k\lambda)$. Solving for $\eta$ gives $\eta \leq C \log^{1/2}(k\lambda\delta_3)T^{-1/2}$.

Therefore, $\mathbb{E}[\mathbf{LCE}_{\widehat{f}}(p_{T':T}, x_{T':T}, y_{T':T})] \leq O(T^{1/2} \log^{1/2}(k\lambda/\delta))$, and when $\mu_1$ and $\mu_2$ are sufficiently separated,

$$\left| \sum_{t = T', \ldots, T} f(g_i \mid x_t) \cdot \mathbb{1}[p_t(x_t) \in v] \cdot (p_t(x_t) - y_t) \right| \leq \widetilde{O}\left(T^{1/2}(\log(1/\delta_2) + \log^{1/2}(k\lambda/\delta))\right)$$

$$+ \widetilde{O}\left(T^{2/3}\sqrt{d} \log^{1/2}(d/\delta_1)\right).$$

Otherwise, without separation, we have

$$\left| \sum_{t = T', \ldots, T} f(g_i \mid x_t) \cdot \mathbb{1}[p_t(x_t) \in v] \cdot (p_t(x_t) - y_t) \right| \leq \widetilde{O}\left(T^{1/2}(\log(1/\delta_2) + \log^{1/2}(k\lambda/\delta))\right)$$

$$+ \widetilde{O}\left(T^{12/13}\sqrt{d} \log^{1/2}(d/\delta_1)\right).$$

**Step 3: Combining Steps 1 and 2.** We now have that with probability $1 - \delta_1$, all parameters are learned within additive error (from step 1); and conditioning on that event, with probability $1 - \delta_2$ that the error incurred in the prediction phase can be bounded as argued in Step 2. We can therefore write the total error incurred over all $T$ samples for any $g_i$ and $v$ when means are separated as, with probability $(1 - \delta_1)(1 - \delta_2) \geq 1 - \delta_1 - \delta_2$,

$$
\left| \sum_{t \in [T]} f(g_i \mid x_t) \cdot \mathbb{1}[p_t(x_t) \in v] \cdot (p_t(x_t) - y_t) \right| \leq \left| \sum_{t \in [T']} f(g_i \mid x_t) \cdot \mathbb{1}[p_t(x_t) \in v] \cdot (p_t(x_t) - y_t) \right|
$$

$$
+ \left| \sum_{t = T', \ldots, T} f(g_i \mid x_t) \cdot \mathbb{1}[p_t(x_t) \in v] \cdot (p_t(x_t) - y_t) \right|
$$

$$
\leq O\left( T^{2/3} \right) + \widetilde{O}\left( T^{2/3} \sqrt{d} \log^{1/2}(d/\delta_1) \right),
$$

and without separation as

$$
\left| \sum_{t \in [T]} f(g_i \mid x_t) \cdot \mathbb{1}[p_t(x_t) \in v] \cdot (p_t(x_t) - y_t) \right| \leq O\left( T^{2/3} \right) + \widetilde{O}\left( T^{12/13} \sqrt{d} \log^{1/2}(d/\delta_1) \right).
$$

The statement of the theorem follows from noting that this holds for any $g_i$ and $v$, and therefore also holds for the maximum $g_i$ and $v$. $\qquad\square$

# C. Supplemental Material for Section 4

## C.1. Algorithms 3 and 4

Here, we give example algorithms that can be used to instantiate a version of the Online Multicalibration Algorithm for Coverable Distinguishers.

Algorithm 3 computes an empirical cover on samples $x_1, \ldots, x_T$, inspired by the classical approach (for binary-valued functions) of Haussler & Welzl (1986).

---
**Algorithm 3** Algorithm for computing a cover
---
1: Input: function class $\mathcal{G} \subset [0,1]^{\mathcal{X}}$, $\varepsilon \in (0,1)$, samples $x_{1:T}$;
2: Initialize empty class $\mathcal{G}'$;
3: For every possible labeling $y_{1:T} \in V_\varepsilon^T$, add to $\mathcal{G}'$ any $g \in \mathcal{G}$ where $g(x_t) \in y_t$ for all $t \in [T]$; $\mathcal{G}'$
---

We also give Algorithm 4, the online multicalibration algorithm of Haghtalab et al. (2023).

---
**Algorithm 4** Online multicalibration algorithm
---
1: Input: $\mathcal{G} \subset [0,1]^{\mathcal{X}}$, $\varepsilon \in (0,1)$, $\lambda, T \in \mathbb{Z}_+$;
2: Initialize Hedge iterate $\ell^{(1)} = \text{Uniform}(\{\pm 1\} \times \mathcal{G} \times V_\lambda)$;
3: **for** $t = 1$ to $T$ **do**
4: $\quad$ Compute $p^{(t)}(x) := \min\limits_{p^*(x) \in \Delta([0, \varepsilon/4\lambda, \ldots, 1])} \max\limits_{y \in [0,1]} \mathbb{E}\limits_{\widehat{y} \sim p^*(x)} \left[ \mathbb{E}\limits_{\ell_{i,g,v} \sim \ell^{(t)}} \left[ i \cdot g(x) \cdot \mathbb{1}[y \in v] \cdot (\widehat{y} - y) \right] \right]$;
5: $\quad$ Announce predictor $p^{(t)}$ to Nature and observe Nature's choice $(x^{(t)}, y^{(t)})$;
6: $\quad$ Update $\ell^{(t+1)} := \text{Hedge}(\widetilde{\ell}^{(1:t)})$ where $\widetilde{\ell}^{(t)}(i, g, v) := 1 - \frac{1}{2} \mathbb{E}\limits_{\widehat{y} \sim p_t(x_t)} \left[ 1 + i \cdot g(x_t) \cdot \mathbb{1}[y_t \in v] \cdot (\widehat{y} - y_t) \right]$;
7: **end for**
---

Algorithm 4 enjoys the following guarantee on multicalibration error.

**Theorem C.1** (Haghtalab et al. (2023) Theorem 4.3). *Fix $\varepsilon > 0$, $\lambda \in \mathbb{Z}_+$, and distinguishers $\mathcal{G} \subset [0,1]^{\mathcal{X}}$. With probability $1 - \delta$, Algorithm 4 guarantees $\mathbf{MCE}(p_{1:T}, x_{1:T}, y_{1:T}; \mathcal{G}) \leq \varepsilon T$ for $T \geq O(\varepsilon^{-2} \ln(|\mathcal{G}| \lambda/\delta))$.*

## C.2. Proof of Theorem 4.4

We prove Theorem 4.4, restated here.

**Theorem 4.4.** *For any real-valued function class $\mathcal{G}$ with finite pseudodimension $\mathrm{Pdim}(\mathcal{G})$, with probability $1 - \delta$, the predictions $p_{1:T}$ made by Algorithm 2 satisfy $\mathbf{MCE}(p_{1:T}, x_{1:T}, y_{1:T}; \mathcal{G}) \leq \widetilde{O}\left(\sqrt{T(\mathrm{Pdim}(\mathcal{G}) + \log(\lambda/\delta))}\right).$*

*Proof of Theorem 4.4.* We will begin by analyzing the multicalibration error incurred on timesteps $T' + 1 \ldots T$. Abusing notation, for any $g \in \mathcal{G}$, let $g' := \arg\inf_{\widetilde{g} \in \mathcal{G}} \frac{1}{T'} \sum_{t \in [T']} |\widetilde{g}(x_t) - g(x_t)|$. Such a $g'$ always exists, because $\mathcal{G}'$ was computed using $x_{1:T'}$. Then, by the triangle inequality, we have that for transcript $H = (p_{T'+1:T}, x_{T'+1:T}, y_{T'+1:T})$:

$$\mathbf{MCE}(H; \mathcal{G}) \leq \underbrace{\sup_{g \in \mathcal{G}} \max_{v \in V_\lambda} \sum_{t \in [T':T]} |g(x_t) - g'(x_t)| \cdot \mathbb{1}[\widehat{y}_t \in v] \cdot |\widehat{y}_t - y_t|}_{(A)} + \underbrace{\mathbf{MCE}(H; \mathcal{G}')}_{(B)}. \tag{9}$$

For $(B)$, $\mathcal{G}$ is of bounded pseudodimension and thus $|\mathcal{G}'| \leq \mathrm{Pdim}(\mathcal{G}) \cdot T^{O(\mathrm{Pdim}(\mathcal{G}))}$ (see Theorem A.1). Standard multicalibration algorithm analysis (Haghtalab et al. (2023) Theorem 4.3; see Theorem C.1) thus gives that with probability $1 - \delta_1$, we have

$$(B) \leq \sqrt{(\mathrm{Pdim}(\mathcal{G})\log(T) + \log(\lambda/\delta_1))(T - T')}.$$

To bound $(A)$, we would like to apply Lemma 4.5. To do so, we will first apply Hoeffding's inequality on the random variables $|g(x_t) - g'(x_t)|$. In particular, note that we can trivially bound $\mathbb{1}[\widehat{y}_t \in v] \cdot |\widehat{y}_t - y_t| \leq 1$ for all $t$; therefore, $(A) \leq \sup_{g \in \mathcal{G}} \sum_{t \in [T':T]} |g(x_t) - g'(x_t)|$, and Hoeffding's inequality gives us that with probability $1 - \delta_2$,

$$\sup_{g \in \mathcal{G}} \sum_{t \in [T':T]} |g(x_t) - g'(x_t)| \leq (T - T') \sup_{g \in \mathcal{G}} \mathbb{E}[|g(x) - g'(x)|] + O(\sqrt{(T - T')\log(1/\delta_2)}).$$

Now, by Lemma 4.5, we have that with probability $1 - \delta_2$,

$$\sup_{g \in \mathcal{G}} \mathbb{E}[|g(x) - g'(x)|] \leq \tfrac{4}{T'}(\mathrm{Pdim}(\mathcal{G})\log(T) + \log(1/\delta_3)).$$

Therefore, with probability $1 - \delta_2 - \delta_3$,

$$\sup_{g \in \mathcal{G}} \sum_{t \in [T':T]} |g(x_t) - g'(x_t)| \leq \tfrac{4(T-T')}{T'}(\mathrm{Pdim}(\mathcal{G})\log(T) + \log(1/\delta_3)) + O(\sqrt{(T - T')\log(1/\delta_2)}).$$

Combining this with (9) and choosing $\delta_1, \delta_2, \delta_3 = \Theta(\delta)$, we have with probability $1 - \delta$,

$$\mathbf{MCE}(p_{T':T}, x_{T':T}, y_{T':T}; \mathcal{G}) \leq O\left(\tfrac{T}{T'}(\mathrm{Pdim}(\mathcal{G})\log(T) + \log(1/\delta)) + \sqrt{T(\mathrm{Pdim}(\mathcal{G})\log(T) + \log(1/\delta))}\right).$$

The claim then follows from noting that $\mathbf{MCE}(p_{1:T'}, x_{1:T'}, y_{1:T'}; \mathcal{G}) \leq T'$ and plugging in the algorithm's choice of $T' = \sqrt{T(\mathrm{Pdim}(\mathcal{G})\log(T) + \log(1/\delta))}$. $\qquad\square$

## C.3. Proof of Lemma 4.5

We prove Lemma 4.5, restated here.

**Lemma 4.5.** *Given $\varepsilon, \delta, T > 0$ and a real-valued function class $\mathcal{G}$ with finite pseudodimension $\mathrm{Pdim}(\mathcal{G})$, consider any $\varepsilon$-cover $\mathcal{G}'$ of $\mathcal{G}$ computed on $T$ randomly sampled datapoints. Then $\mathcal{G}'$ is a $4\varepsilon$-cover of $\mathcal{G}$ with probability at least $1 - O\left(\mathrm{Pdim}(\mathcal{G})^2 (1/\varepsilon)^{O(\mathrm{Pdim}(\mathcal{G}))} \exp(-T\varepsilon)\right).$*

Lemma 4.5 follows from the following more general presentation stated for any real-valued function class whose covering number grows at a sub-square-root-exponential rate in the number of datapoints. The result follows a similar approach to Haussler & Welzl (1986).

**Lemma C.2.** *Given $\varepsilon, \delta, T > 0$ and a real-valued function class $\mathcal{G}$ where $N_1(\varepsilon/96, \mathcal{G}, 2T) \leq \sqrt{\frac{1}{8} \exp(T\varepsilon^2/32)}$, consider any $\varepsilon$-cover $\mathcal{G}'$ of $\mathcal{G}$ computed on $T$ random datapoints. Then $\mathcal{G}'$ is a $4\varepsilon$-cover of $\mathcal{G}$ with probability at least $1 - O(N_1\left(\frac{\varepsilon}{8}, \mathcal{G}, 2T\right)^2 \exp(-T\varepsilon))$.*

Given Lemma C.2, we can prove Lemma 4.5 fairly directly.

**Lemma 4.5.** *Given $\varepsilon, \delta, T > 0$ and a real-valued function class $\mathcal{G}$ with finite pseudodimension $\mathrm{Pdim}(\mathcal{G})$, consider any $\varepsilon$-cover $\mathcal{G}'$ of $\mathcal{G}$ computed on $T$ randomly sampled datapoints. Then $\mathcal{G}'$ is a $4\varepsilon$-cover of $\mathcal{G}$ with probability at least $1 - O\big(\mathrm{Pdim}(\mathcal{G})^2 (1/\varepsilon)^{O(\mathrm{Pdim}(\mathcal{G}))} \exp(-T\varepsilon)\big)$.*

*Proof.* Given that $\mathcal{G}$ is of bounded pseudodimension, we have that the covering number is bounded by $N_1(\varepsilon, \mathcal{G}, T) \leq O\left(\mathrm{Pdim}(\mathcal{G})(1/\varepsilon)^{O(\mathrm{Pdim}(\mathcal{G}))}\right)$ (Anthony & Bartlett (1999) Thm. 18.4; see Fact A.1). We also can observe that our upper bound on $N_1(\varepsilon, \mathcal{G}, T)$ is independent of $T$. Thus, we can apply Lemma C.2 to note that $\mathcal{G}'$ is a $4\varepsilon$-cover of $\mathcal{G}$ with probability at least $1 - O(\mathrm{Pdim}(\mathcal{G})^2 (1/\varepsilon)^{O(\mathrm{Pdim}(\mathcal{G}))} \exp(-T\varepsilon))$.

$\square$

To prove Lemma C.2, we first recall the following one-sided testing form of Bernstein's inequality.

*Fact* C.3. Let $X_1, \ldots, X_T$ be i.i.d. random variables supported on $[0, 1]$ with mean $\mu = \mathbb{E}[X_i]$, and let $\widehat{\mu} = \frac{1}{T} \sum_{i=1}^{T} X_i$ be the sample mean. Then, for any $\varepsilon > 0$, we have:

$$\text{If } \mu > 3\varepsilon, \text{ then } \Pr(\widehat{\mu} \leq 2\varepsilon) \leq \exp\left(-\frac{T\varepsilon}{8}\right).$$

*Proof of Fact C.3.* Applying Bernstein's inequality with deviation $t = \mu - 2\varepsilon \geq \mu/3 \geq \varepsilon$ gives:

$$\Pr(\widehat{\mu} \leq 2\varepsilon) = \Pr(\mu - \widehat{\mu} \geq t) \leq \exp\left(\frac{-Tt^2}{2(\mu + \frac{t}{3})}\right) \leq \exp\left(\frac{-\frac{1}{3}T\mu\varepsilon}{2(\frac{4}{3}\mu - \frac{2}{3}\varepsilon)}\right) = \exp\left(\frac{-T\mu\varepsilon}{4(2\mu - \varepsilon)}\right) \leq \exp\left(-\frac{T\varepsilon}{8}\right).$$

$\square$

We next proceed to the main proof.

*Proof of Lemma C.2.* **Existence of covering points.**

First, note that for any $\varepsilon$ and $m$, $N_1(\varepsilon, \{f(x) - g(x) \mid f, g \in \mathcal{G}\}, m) \leq N_1(\varepsilon, \mathcal{G}, m)^2$. Then, recalling that bounded covering number implies uniform convergence,

$$\Pr\left(\sup_{f,g \in \mathcal{G}} \mathbb{E}\left[|f(x) - g(x)|\right] - \frac{1}{T} \sum_{t=1}^{T} |f(x_t') - g(x_t')| \geq \varepsilon/8\right) \leq 4N_1(\varepsilon/(16 \cdot 8), \mathcal{G}, 2T)^2 \exp\left(-T\varepsilon^2/32\right).$$

Since $8N_1(\varepsilon/(16 \cdot 8), \mathcal{G}, 2T)^2 \leq \exp\left(T\varepsilon^2/32\right)$:

$$\Pr\left(\sup_{f,g \in \mathcal{G}} \mathbb{E}\left[|f(x) - g(x)|\right] - \frac{1}{T'} \sum_{t=1}^{T'} |f(x_t') - g(x_t')| \geq \varepsilon/8\right) \leq \frac{1}{2}.$$

Thus, by the probabilistic method, there exists some sequence of datapoints $x_1, \ldots, x_T'$ such that

$$\sup_{f,g \in \mathcal{G}} \mathbb{E}\left[|f(x) - g(x)|\right] - \frac{1}{T} \sum_{t=1}^{T} |f(x_t') - g(x_t')| \leq \varepsilon/8. \tag{10}$$

**Covering failure events.** Let $x_1, \ldots, x_T$ denote i.i.d. samples from distribution $\mathcal{D}$. By definition, there is a subset $\mathcal{G}' \subset \mathcal{G}$ of size $N_1(\varepsilon, \mathcal{G}, T)$ such that for all $f \in \mathcal{G}$, there is a $f' \in \mathcal{G}'$ such that $\frac{1}{T} \sum_{t=1}^{T} |f(x_t) - f'(x_t)| \leq \varepsilon$. Note that this subset $\mathcal{G}'$ is not dependent on $\mathcal{D}$ and can be computed from $x_1, \ldots, x_T$.

We next define, for any $f, g \in \mathcal{G}$, the event $E_{f,g}$ to be the event that both $\mathbb{E}\left[|f(x) - g(x)|\right] \geq 4\varepsilon$ and $\frac{1}{T}\sum_{t=1}^{T}|f(x_t) - g(x_t)| \leq \varepsilon$.

Condition on none of the events in $\{E_{f,g} \mid f, g \in \mathcal{G}\}$ occurring. Fix any $f \in \mathcal{G}$. Since we covered $\mathcal{G}$ on $x_1, \ldots, x_T$ with a tolerance of $\varepsilon$, there is a $g \in \mathcal{G}'$ such that $\frac{1}{T}\sum_{t=1}^{T}|f(x_t) - g(x_t)| \leq \varepsilon$. Since $E_{f,g}$ did not occur, we know that $\mathbb{E}\left[|f(x) - g(x)|\right] \leq 4\varepsilon$. This implies that $\mathcal{G}'$ is a $4\varepsilon$-net as desired.

It thus suffices to upper bound $\Pr\left[\bigcup_{f,g\in\mathcal{G}} E_{f,g}\right]$.

**Bounding $\bigcup \Pr[E_{f,g}]$ by covering $\mathcal{G}$.** We now define, for any $f, g \in \mathcal{G}$, the event $\widetilde{E}_{f,g}$ to be the event that both $\mathbb{E}\left[|f(x) - g(x)|\right] \geq 3\varepsilon$ and $\frac{1}{T}\sum_{t=1}^{T}|f(x_t) - g(x_t)| \leq 2\varepsilon$.

Let $\widehat{\mathcal{G}}$ be a $\left(\frac{\varepsilon}{4}\right)$-covering of $\mathcal{G}$ on the datapoints $x'_1, \ldots, x'_T, [x_1, \ldots, x_T]_M$. Therefore, $\widehat{\mathcal{G}}$ is a cover of size $N_1(\frac{\varepsilon}{8}, \mathcal{G}, 2T)$. This means for every $f \in \mathcal{G}$ there is a $\widehat{f} \in \widehat{\mathcal{G}}$ such that

$$\frac{1}{2T}\left(\sum_{t=1}^{T}\left|f(x'_t) - \widehat{f}(x'_t)\right| + \sum_{t=1}^{T}\left|f(x_t) - \widehat{f}(x_t)\right|\right) \leq \frac{\varepsilon}{8}.$$

We now proceed to show that $\bigcup_{f,g\in\mathcal{G}} E_{f,g} \subseteq \bigcup_{\widehat{f},\widehat{g}\in\widehat{\mathcal{G}}} E_{\widehat{f},\widehat{g}}$.

First, by the triangle inequality,

$$\frac{1}{T}\sum_{t=1}^{T}\left|\widehat{f}(x_t) - \widehat{g}(x_t)\right| \leq \frac{1}{T}\sum_{t=1}^{T}|f(x_t) - g(x_t)| + \left|\frac{1}{T}\sum_{t=1}^{T}|f(x_t) - g(x_t)| - \frac{1}{T}\sum_{t=1}^{T}\left|\widehat{f}(x_t) - \widehat{g}(x_t)\right|\right|$$
$$\leq \frac{1}{T}\sum_{t=1}^{T}|f(x_t) - g(x_t)| + \varepsilon.$$

Therefore, $\{\frac{1}{T}\sum_{t=1}^{T}|f(x_t) - g(x_t)| \leq \varepsilon\}$ only if $\{\frac{1}{T}\sum_{t=1}^{T}\left|\widehat{f}(x_t) - \widehat{g}(x_t)\right| \leq 2\varepsilon.\}$

Next, by the definition of $\widehat{\mathcal{G}}$, we have

$$\frac{1}{T}\sum_{t=1}^{T}\left|f(x'_t) - \widehat{f}(x'_t)\right| \leq \varepsilon/4 \quad \text{and} \quad \frac{1}{T}\sum_{t=1}^{T}\left|f(x_t) - \widehat{f}(x_t)\right| \leq \varepsilon/4.$$

By the triangle inequality, for every $f, g \in \mathcal{G}$ and their matching $\widehat{f}, \widehat{g} \in \widehat{\mathcal{G}}$, we have

$$\left|\frac{1}{T}\sum_{t=1}^{T}|f(x'_t) - g(x'_t)| - \frac{1}{T}\sum_{t=1}^{T}\left|\widehat{f}(x'_t) - \widehat{g}(x'_t)\right|\right| \leq \frac{\varepsilon}{2} \quad \text{and} \quad \left|\frac{1}{T}\sum_{t=1}^{T}|f(x_t) - g(x_t)| - \frac{1}{T}\sum_{t=1}^{T}\left|\widehat{f}(x_t) - \widehat{g}(x_t)\right|\right| \leq \frac{\varepsilon}{2}.$$

$$(11)$$

Thus, repeatedly applying the triangle inequality (first, third, and fifth transitions below),

$$
\begin{aligned}
\mathbb{E}\left[|f(x) - g(x)|\right] &\leq \tfrac{1}{T}\sum_{t=1}^{T}|f(x_t') - g(x_t')| + \left|\mathbb{E}\left[|f(x) - g(x)|\right] - \tfrac{1}{T}\sum_{t=1}^{T}|f(x_t') - g(x_t')|\right| \\
&\leq \tfrac{1}{T}\sum_{t=1}^{T}|f(x_t') - g(x_t')| + \tfrac{\varepsilon}{4} \\
&\leq \tfrac{1}{T}\sum_{t=1}^{T}\left|\widehat{f}(x_t') - \widehat{g}(x_t')\right| + \left|\tfrac{1}{T}\sum_{t=1}^{T}|f(x_t') - g(x_t')| - \tfrac{1}{T}\sum_{t=1}^{T}\left|\widehat{f}(x_t') - \widehat{g}(x_t')\right|\right| + \tfrac{\varepsilon}{4} \\
&\leq \tfrac{1}{T}\sum_{t=1}^{T}\left|\widehat{f}(x_t') - \widehat{g}(x_t')\right| + \tfrac{3\varepsilon}{4} \\
&\leq \mathbb{E}\left[\left|\widehat{f}(x) - \widehat{g}(x)\right|\right] + \left|\mathbb{E}\left[\left|\widehat{f}(x) - \widehat{g}(x)\right|\right] - \tfrac{1}{T}\sum_{t=1}^{T}\left|\widehat{f}(x_t') - \widehat{g}(x_t')\right|\right| + \tfrac{3\varepsilon}{4} \\
&\leq \varepsilon + \mathbb{E}\left[\left|\widehat{f}(x) - \widehat{g}(x)\right|\right],
\end{aligned}
$$

where the second and final transitions are due to (10) and the fourth is due to (11).

Thus, for any $f, g \in \mathcal{G}$, $\mathbb{E}\left[|f(x) - g(x)|\right] \geq 4\varepsilon$ only if the corresponding $\widehat{f}, \widehat{g} \in \mathcal{G}$ satisfies $\mathbb{E}\left[\left|\widehat{f}(x) - \widehat{g}(x)\right|\right] \geq 3\varepsilon$. It follows that $\Pr\left(\bigcup_{f,g \in \mathcal{G}} E_{f,g}\right) \leq \Pr\left(\bigcup_{\widehat{f},\widehat{g} \in \widehat{\mathcal{G}}} \widetilde{E}_{\widehat{f},\widehat{g}}\right)$.

Finally, for any fixed choice of $f$ and $g$, we have by Fact C.3 that $\Pr\left[\widetilde{E}_{f,g}\right] \leq \exp\left(-\tfrac{T\varepsilon}{8}\right)$. Thus, union bounding over all pairs $\widehat{f}, \widehat{g} \in \widehat{\mathcal{G}}$, we have

$$
\Pr\left(\bigcup_{f,g \in \mathcal{G}} \widetilde{E}_{f,g}\right) \leq \left|\widehat{\mathcal{G}}\right|^2 \exp\left(-\tfrac{T\varepsilon}{8}\right) \leq O(N_1(\tfrac{\varepsilon}{8}, \mathcal{G}, 2T)^2 \exp(-T\varepsilon)).
$$

$\square$

### C.4. Proofs of pseudodimension bounds (Lemmas 4.6 and 4.7

We begin by proving Lemma 4.6, restated below.

**Lemma 4.6.** *Let $D := \mathrm{Pdim}(\mathcal{H})$ where $\mathcal{H}$ is defined as in (3). Then, the binary-valued distinguisher class $\mathcal{G} = \{x \mapsto \mathbb{1}[g = \arg\max_{g' \in [k]} \widetilde{f}(g' \mid x)] \mid g \in [k], \widetilde{f} \in \mathcal{F}\}$ has pseudodimension (equivalently VC dimension) of at most $O(kD\log(k))$, and the real-valued distinguisher class $\mathcal{G}' = \{x \mapsto \widetilde{f}(g \mid x) \mid g \in [k], \widetilde{f} \in \mathcal{F}\}$ has pseudodimension at most $O(kD\log^2(kD))$.*

*Proof.* For the first claim, consider the class $\widetilde{\mathcal{G}}$ that consists of comparisons of two groups' likelihoods, rather than an argmax over all $k$ groups as in $\mathcal{G}$:

$$
\widetilde{\mathcal{G}} := \{x \mapsto \mathbb{1}[\widetilde{f}(x \mid g) > \widetilde{f}(x \mid g')] \mid g, g' \in [k], \widetilde{f} \in \mathcal{F}\}.
$$

We can verify that $\widetilde{\mathcal{G}} \subseteq \{x \mapsto \mathbb{1}[h(x) > 1] \mid h \in \mathcal{H}\}$. Note that $\mathrm{VC}(\{x \mapsto \mathbb{1}[h(x) > 1] \mid h \in \mathcal{H}\}) = \mathrm{Pdim}(\mathcal{H})$ by definition. Since the $k$-fold intersection of a VC class increases VC dimension by at most a factor of $k\log(k)$ (Blumer et al. (1989), Lemma 3.2.3; see Fact A.6), we have $\mathrm{VC}(\{x \mapsto \mathbb{1}[h_1(x) > 1 \wedge \ldots h_k(x)] \mid h_1, \ldots, h_k \in \mathcal{H}\}) \leq O(kD\log(k))$. Finally, we can verify that

$$
\mathcal{G} \subseteq \{x \mapsto \mathbb{1}[h_1(x) > 1 \wedge \ldots h_k(x)] \mid h_1, \ldots, h_k \in \mathcal{H}\}, \tag{12}
$$

concluding our first claim.

For the second claim, fix $g = 1$ without loss of generality. Note that for any $f \in \mathcal{F}$, we can write $f(1 \mid x)$ as

$$f(1 \mid x) = \frac{w_1 \cdot f(x \mid 1)}{\sum_{j \in [k]} w_j \cdot f(x \mid j)} = \frac{1}{\sum_{j \in [k]} \frac{w_1 f(x|1)}{w_j f(x|j)}} = \frac{1}{\sum_{j \in [k]} \frac{f(1|x)}{f(j|x)}}.$$

Thus, $\mathrm{Pdim}(\mathcal{G}') = \mathrm{Pdim}\left(\left\{ x \mapsto \frac{1}{\sum_{j \in [k]} \frac{f(1|x)}{f(j|x)}} \mid f \in \mathcal{F} \right\}\right) = \mathrm{Pdim}\left(\left\{ x \mapsto \frac{1}{\sum_{j \in [k]} h(x)} \mid h \in \mathcal{H} \right\}\right).$

The sum of bounded pseudodimension classes enjoy an almost linear bound in pseudodimension (Attias & Kontorovich (2024) Theorem 1; see Fact A.4), so $\mathrm{Pdim}\left(\left\{ x \mapsto \sum_{j \in [k]} h_j \mid h_1, \ldots, h_k \in \mathcal{H} \right\}\right) \leq kD \log^2(kD)$. Finally, $x \mapsto 1/x$ is monotonic, and therefore preserves pseudodimension (Fact A.3). $\qquad \square$

Finally, we prove Lemma 4.7.

**Lemma 4.7.** *When $\mathcal{F}$ is an exponential family (Definition 2.4), the pseudodimension of $\mathcal{H}$ (with $\mathcal{H}$ defined as in (3)) is bounded by the dimension of the sufficient statistic, i.e. $\mathrm{Pdim}(\mathcal{H}) \leq \dim(T(x)) + 1$.*

*Proof of Lemma 4.7.* Because $\mathcal{F}$ is an exponential family, we can write

$$\frac{f(1 \mid x)}{f(j \mid x)} = \frac{w_1 \cdot f(x \mid 1)}{w_j \cdot f(x \mid j)} = \frac{w_1 \cdot g(\theta_1) h(x) \exp(\langle \theta_1, T(x) \rangle)}{w_j \cdot g(\theta_j) h(x) \exp(\langle \theta_j, T(x) \rangle)} = \frac{w_1 \cdot g(\theta_1)}{w_j \cdot g(\theta_j)} \exp(\langle \theta_1 - \theta_j, T(x) \rangle).$$

Note that $\langle \theta_1 - \theta_j, T(x) \rangle$ is a vector space of size $\dim(T(x))$; by Fact A.2, $\mathrm{Pdim}(\langle \theta_1 - \theta_j, T(x) \rangle) \leq \dim(T(x)) + 1$. The claim follows from noting that $\exp(\cdot)$ is monotonic and thus preserves pseudodimension (Fact A.3). $\qquad \square$

