# OpenReview forum: "Learning With Multi-Group Guarantees For Clusterable Subpopulations"
_ICML.cc/2025/Conference — ICML 2025 poster_

### Official Review · Reviewer_U4DR · 2025-03-10

**Overall Recommendation:** 4

**Summary:**

This paper focuses on providing *multigroup* guarantees (with a focus on multicalibration, though the techniques are more general) in a stochastic online prediction game in the novel setting where the groups of interest are not provided beforehand as functions of the feature values, but, rather, as unknown endogeneous subpopulations that emerge from the distribution of individuals. This provides an alternative view of recent "multigroup" guarantees that has an unsupervised flavor; instead of *predefining* a collection of groups (typically a collection of indicator functions on the input space), this paper seeks to argue that another perspective is where groups arise naturally from the true distribution generating the input individuals.

The authors provide two main algorithms/approaches for providing multi-group calibration guarantees for this model. The first "warmup algorithm," denoted "Cluster-then-Predict," provides suboptimal $O(T^{2/3})$ rates under rather restrictive assumptions. The algorithm is simple: (1) take some tuned number of timesteps to learn the underlying group clusters and then (2) run a marginal calibration algorithm for each group for the remaining timesteps. The second algorithm uses a multicalibration approach to provide a better $O(T^{1/2})$ rate under more general assumptions (that the clusters are drawn from exponential families). The key algorithmic idea here is to use multicalibration on a pre-defined collection of groups/distinguishers; specifically, the collection that is defined by the possible family of density ratios under consideration. Because this collection is not too large combinatorially, the main algorithm proceeds by (i) constructing a cover and then (ii) running a multicalibration algorithm over the cover.

**Claims And Evidence:**

Yes, I believe all the claims are supported with clear and convincing evidence. The main claims are the guarantees for the "Cluster-then-Predict" warmup algorithm, in **Proposition 3.1** and **Proposition 3.4**, and the guarantee for the cover + multicalibrate algorithm in **Theorem 4.1.** I have verified the proofs for all these claims, and the proofs are correct to my verification.

An additional, more normative claim of the paper is present in Appendix A, which provides an argument for the endogeneous subgroup model of the paper. I found this argument persuasive (and refreshing! I believe more papers in this space should grapple with the implications of their mathematical assumptions), particularly in that this model "contextualizes" individuals with others whom predictions are being made. The authors make an interesting distinction in this Appendix article on how their model differs from the model accepted in most multicalibration/multigroup learning literature -- rather than viewing subgroup membership as "computationally-identifiable" groups, this clustering viewpoint views subgroups as "statistically identifiable."

**Essential References Not Discussed:**

I believe that the authors have cited all relevant work. It doesn't seem to me that there are any references missing (though I am less familiar with the unsupervised learning/clustering literature).

**Experimental Designs Or Analyses:**

This is a theory paper, so no experiments to check.

**Methods And Evaluation Criteria:**

Yes, the main methods and evaluation criteria were proofs that the authors' two main desiderata (denoted as Discriminant Calibration Error and Likelihood Calibration Error) indeed decay sublinearly in the stochastic online model for both their algorithms. The desiderata themselves are natural and are well-accepted notions of prediction quality in the multicalibration literature.

In particular, DCE measures the $\ell_{\infty}$ norm of the calibration error over all groups on all the rounds for which a particular group is the "most likely." LCE measures the $\ell_{\infty}$ norm of the calibration error over all groups, but the rounds are weighted according to the generative model for the groups. These are the natural definitions of prediction quality that arise if one is concerned about the model the authors pose for their problem (the *endogeneous subgroups generative model*) and calibration as a measure of prediction quality.

**Other Comments Or Suggestions:**

Here are some other minor suggestions that might improve the presentation of the paper:

- Page 4: I would suggest introducing the Section in the paragraph before **Minimizing discriminant calibration error...** with a disclaimer and motivation that the section only deals with two groups that are Gaussian. I understand that it is a warmup, but it may slightly clarify the presentation to introduce the section as dealing with this special case first.
- Page 5: A nitpick for Propositions 3.1 and 3.4 is to define $d$ in the prop statements.
- Page 6: It may clarify presentation for readers less familiar with online prediction to mention that $O(T^{1/2})$ is often minimax optimal.

**Other Strengths And Weaknesses:**

**Strengths**
- The work is *extremely* well-written and clear. I had no trouble following the arguments, the overall flow of the paper, and the claims and model are well-motivated. The paper was a pleasure to read.
- This alternative view of subpopulations is an interesting and novel alternative to the typical view of group structure in this literature, and I believe that it is quite worth disseminating. In particular, the model is natural, and I believe that it should be further studied.
- The algorithmic results are interesting -- I particularly liked the key idea of cleverly choosing the class of distinguishers for multicalibration to be the class of possible density ratios, which are combinatorially bounded. This highlights how far multicalibration algorithms can go with a well-defined class of distinguishers for the problem at hand.
- Though "fuzzier," I believe that this a key strength that should not be overlooked is the focus the authors give on the model and the normative assumptions underlying the model. Reading Appendix A was a pleasure, and it is rare to find authors giving the requisite thought to the assumptions underlying their model/problem of choice. I believe that this is particularly lacking (though very warranted) in subfields where the algorithms concern predictions with a human dimension, where this paper is clearly situated.

**Weaknesses*
The following are more questions than glaring weaknesses. The following contributions might make the paper more complete, but I believe that their omission does not detract from the overall quality of the paper (they seem to me to be more "next steps" or ways to round out the story).

- This is not a huge weakness, but it might make the paper more complete to have a corresponding batch setup, with results for the batch case. I wonder if the authors have already considered this and ran into a snag with, say, a boilerplate online-to-batch conversion of their algorithm? I imagine this is straightforward, though I might be missing something.
- The rates presented all depend on $T$ indiscriminately for *every* group, which I understand is part of the definition of DCE and LCE (as both definitions take a max over the groups). However, is it possible to obtain more fine-grained guarantees for each group for DCE corresponding to the number of rounds for which the group was active? If $T_g := \sum_{t = 1}^T \mathbf{1}\{ g = \mathrm{argmax}_{j \in [k]} f( j \mid x_t, y_t) \}$, is it possible to achieve a more fine-grained bound where DCE depends on $T_g$ for each group?

**Questions For Authors:**

See "Weaknesses" above. I will just paste the same questions I had above here:

1. This is not a huge weakness, but it might make the paper more complete to have a corresponding batch setup, with results for the batch case. I wonder if the authors have already considered this and ran into a snag with, say, a boilerplate online-to-batch conversion of their algorithm? I imagine this is straightforward, though I might be missing something.
2. The rates presented all depend on $T$ indiscriminately for *every* group, which I understand is part of the definition of DCE and LCE (as both definitions take a max over the groups). However, is it possible to obtain more fine-grained guarantees for each group for DCE corresponding to the number of rounds for which the group was active? If $T_g := \sum_{t = 1}^T \mathbf{1}\{ g = \mathrm{argmax}_{j \in [k]} f( j \mid x_t, y_t) \}$, is it possible to achieve a more fine-grained bound where DCE depends on $T_g$ for each group?

**Relation To Broader Scientific Literature:**

This work can be situated squarely in the literature on multicalibration initialized by [HKRR18], and, more broadly, the literature concerned with obtaining multigroup guarantees over computationally-identifiable subpopulations. The authors provide a different perspective, however, than the model where groups are computationally-identifiable collections of indicator functions (typically finite or in a VC class). Instead, they consider clusterable groups coming from a natural generative model. The main tool used in this work draws from an algorithm of [HJZ23] instantiated with an appropriate collection of groups/distinguishers. The work also touches on clustering and unsupervised learning; their first algorithm relies on black-box access to a clustering algorithm, specifically the algorithm for Gaussian clustering of [AAW13].

[HKRR18] Hébert-Johnson, Ursula, et al. "Multicalibration: Calibration for the (computationally-identifiable) masses." International Conference on Machine Learning. PMLR, 2018.
[HJZ23] Nika Haghtalab, Michael Jordan, and Eric Zhao. "A unifying perspective on multi-calibration: Game dynamics for multi-objective learning." Advances in Neural Information Processing Systems 36 (2023): 72464-72506.
[AAW13] Azizyan, Martin, Aarti Singh, and Larry Wasserman. "Minimax theory for high-dimensional gaussian mixtures with sparse mean separation." Advances in Neural Information Processing Systems 26 (2013).

**Theoretical Claims:**

I checked all the main claims for the algorithms' correctness (Prop 3.1, Prop 3.4, Theorem 4.1 and its corresponding lemmas). I carefully went through the proof sketches or proofs that were in the main body, and I read over the Appendix C and D arguments, though less carefully than when I checked claims in the main body. These theoretical claims all seem to check out.

---

> ### Author Rebuttal · Authors · 2025-04-01
>
> Thanks for your detailed review and suggestions! To briefly address your questions:
>
> **Q1: Online-to-batch.**
>
> > This is not a huge weakness, but it might make the paper more complete to have a corresponding batch setup, with results for the batch case. I wonder if the authors have already considered this and ran into a snag with, say, a boilerplate online-to-batch conversion of their algorithm?
>
> Yes, the boilerplate online-to-batch conversion works fine to extend our guarantees to the batch setting. Indeed in  the batch setting, even an algorithm that is similar in style to our cluster-then-predict is slightly less egregious. That is, because there would be no “explore-then-commit” penalty—all $T$ points could be reused to first learn the likelihoods $f(g | x)$ and then learn to make predictions $f (y | x)$ )—achieves near-optimal rate in T but still suffers from a dependence on the separation between clusters ($\gamma$). Our multi-objective algorithm entirely avoids the dependence as well. We agree this is an interesting point worth making, especially since it already follows directly from our online result—we will add a discussion on this.
>
> **Q2: Group-dependent rates.**
>
> > The rates presented all depend on T indiscriminately for every group… Is it possible to obtain more fine-grained guarantees for each group for DCE corresponding to the number of rounds for which the group was active?
>
> This is an interesting question! Group-dependent rates are indeed attainable through our reduction to multicalibration, where one can attain rates that scale with $\sqrt{T \Pr(x \in g)}$ “for free” (see e.g. Theorem 3.1 [1], Theorem 5.7 in [2]). This is in a sense optimal; the right concentration rate for a group of probability mass P is $1/\sqrt{T P}$. Group-dependent rates are not the focus of our paper, but we agree it will be nice to highlight that it comes for free with our reduction.
>
> [1] Noarav et.al. High-Dimensional Prediction for Sequential Decision Making
>
> [2] Haghtalab et.al. A Unifying Perspective on Multicalibration: Game Dynamics for Multi-Objective Learning

---

### Official Review · Reviewer_iw9y · 2025-03-12

**Overall Recommendation:** 4

**Summary:**

The paper considers a multi-group online learning problem with instances $(x_t, y_t)$ arriving in sequence. In contrast to prior work in online multi-group learning, the groups themselves are not known at each instance. Instead, the paper assumes that there is an endogenous unknown subgroup model, such as, e.g.,  a mixture of Gaussians. We can think of this model as first sampling a mixture component (based on a distribution components), then sampling an $x_t$ from this mixture, and then finally sampling a label from some unknown distribution $f(y_t | x_t)$. At each time step $t$, the learner should produce some action $a_t$.

Given a datapoint $x_t$, it is impossible to identify which “group” (or mixture component) generated it. Therefore, two types of errors are considered. The first, discriminant error, takes into account only the most likely group that $x_t$ belongs to. The second, likelihood error, weighs each group that $x_t$ could have been generated from by the likelihood that it was generated by that group (mixture component).

The main concern of the paper is to generate calibrated predictions which have bounded worst group discriminant and likelihood error. First, the authors utilize the common technique of discretizing the outputs of the model into “buckets” when considering calibration errors. Then, they introduce two calibration variants to the aforementioned errors: discriminant calibration error (DCE) and likelihood calibration error (LCE). DCE can be thought of as minimizing the discriminant error over the worst “group intersect calibration bucket” in the sequence $x_t, y_t$. That is, for each mixture component and each prediction bucket (for example all predictions in $[0, 0.2]$ or $[0.2, 0.4]$), sum up the error of all predictions / actions taken by the learner in the sequence which belong to that group and that bucket. “Belong” here is defined in the discriminant way. LCE is the identical concept, but “Belong” is instead a probabilistic notion, since one $x_t$ can contribute to the error of multiple groups (depending on the probability of that “group” / mixture component generating $x_t$).

With DCE and LCE in hand, the paper then proposes some algorithms to achieve bounded error. To start with, a standard “cluster-then-predict” algorithm is proposed for Gaussian mixtures. This algorithm first learns the underlying groups (mixture components), then learns calibrated predictors for each group independently (a la Foster and Vohra.). The paper shows that this family of algorithms can achieve a $O(T^{⅔})$ error rate on DCE and LCE, and indeed, that this is tight due to the difficulty in determining underlying mixtures. In addition, the bound has a necessary dependence on a separation parameter $\gamma$ for the underlying mixtures.

The paper then considers a new family of algorithms which _do not_ actually learn the underlying groups. In particular, if one can provide multicalibrated predictions for a suitable cover of all possible groups, this should be sufficient to obtain good worst-group performance guarantees. The main result of the paper, Theorem 4.1, shows that such an approach is feasible and enjoys the better DCE/LCE error rate of $O(\sqrt{T})$ over the cluster-then-predict family of algorithms. Furthermore, no “separation” parameter dependence on $\gamma$ is necessary. This new family of algorithms works by efficiently multicalibrating over a finite cover of the likelihood ratios of the underlying density class.

**Claims And Evidence:**

Yes, all proofs are present.

**Essential References Not Discussed:**

Given the technique of multicalibrating w.r.t. likelihood ratios of the underlying function class, it may be useful to discuss [1], which study the problem of (offline) learning multicalibrated partitions via a likelihood ratio / importance weight approach. I am not an expert in this area, but at a surface level the technique seems to be similar. However, in the submitted paper, concrete bounds on the pseudo-dimension of the derived likelihood ratio are discussed and utilized, whereas this doesn’t seem to be discussed in [1].

[1]: Multicalibrated Partitions for Importance Weights. Gopalan et al. 2021.

**Experimental Designs Or Analyses:**

N/A

**Methods And Evaluation Criteria:**

N/A

**Other Comments Or Suggestions:**

I enjoyed the extended discussion in Appendix A. Defining subpopulations via statistical identifiability. I think this discussion represents an important dilemma in the multicalibration literature, namely that groups are known / deterministic. Algorithms like the proposed one which allow for partial / probabilistic and _unknown_ group membership may represent an important development in the literature. I would suggest this discussion be included in the main paper somehow (or at least an abridged version of it), perhaps possible with the increased final paper page count.

**Other Strengths And Weaknesses:**

I think this is a technically strong paper on an interesting (and to my knowledge, previously unexplored) problem: online multicalibration with unknown groups.

I would suggest increasing the amount of discussion and removing most proofs from the main paper. For example, the proof of lemma 4.5 is a fairly straightforward argument to bound the pseudodimension of a derived function class — this could be replaced with a more detailed proof sketch of Theorem 4.4..

**Questions For Authors:**

1. Since $\mathcal{F}$ is the underlying density class with bounded pseudo-dimension, I understand that we can bound the pseudo dimension of $\mathcal{H}$, the class of density ratios on $\mathcal{F}$. To obtain theorem 4.4, we need to run algorithm 2 with $\mathcal{G} = \mathcal{H}$, correct? So when we are computing our approximate cover (a la appendix D.2), I should think of computing a cover on the likelihood ratios of the original function class?

2. Is there a natural, intuitive interpretation of DCE / LCE and the difference between the two? Maybe a simple two cluster example may help distinguish what the different error rates may signify?

3. Is the proposed algorithm 2 computationally tractable?

**Relation To Broader Scientific Literature:**

This paper relates to a line of work on online multicalibration. Importantly, the paper does not assume that group membership is known (or even that it is deterministic). To the best of my knowledge, this sets it apart from previous work in the area (although I do not work in online multicalibration, and may not be totally up to date with the literature).

**Theoretical Claims:**

I did not check the correctness of any proof. Nonetheless, the main technical insight seems to be a combination of the following two facts 1) we should apply online multicalibration algorithms with the “right” groups (here, using $\mathcal{H}$, the likelihood ratios of the underlying density function class); and 2) An approximate covering can be computed efficiently in the first phase of the algorithm.

Both facts seem reasonable as someone who has not checked the proofs carefully. Fact 2) is especially interesting, since one can apparently learn a good covering with less data than one can actually identify the underlying groups (see, for example, the similarity between phase 1 of algorithm 1 (cluster-then-predict) and algorithm 2 (online multicalibration approach)).

---

> ### Author Rebuttal · Authors · 2025-04-01
>
> Thanks for your review and suggestions! We will incorporate your presentation recommendations in the next versions.
>
> **Q1: Covering of H**
>
> > To obtain theorem 4.4, we need to run algorithm 2 with G=H, correct?
>
> For Theorem 4.4, we run Algorithm 2 with $\mathcal{G}$ defined according to eq. 1 (for DCE) or eq. 2 (for LCE) — essentially, $\mathcal{G}$ corresponds to whatever the proper notion of “group membership” should be (for either DCE/LCE). Thus, the cover computed in the first phase of Algorithm 2 is a cover on the function classes corresponding to the group membership functions, but it turns out that the size of those covers only needs to depend on the pseudodimension of the class of likelihood ratios.
>
> **Q2: DCE vs LCE**
>
> > Is there a natural, intuitive interpretation of DCE / LCE and the difference between the two?
>
> The difference between DCE and LCE as (per-subgroup) performance measures is critical when subgroups are overlapping.
>
> As one example, first consider two well-separated standard Gaussians supported on a number line; for example, suppose their centers are at x=-1000 and x=1000 respectively. For any given candidate predictor, LCE and DCE will be nearly identical since $f(g_1 \mid x) \approx 1[ f(g_1 \mid x) > f(g_2 \mid x)$. In particular, both LCE and DCE can be understood as enforcing that a predictor is calibrated on the part of the data distribution where x<0 and calibrated on the part where x>0.
>
> Now suppose we reduced the separation between the two Gaussians so that their centers are x=-0.01 and x=0.01 respectively. LCE and DCE will now differ significantly. LCE measures calibration error on two groups whose likelihood ratio is ~50/50 across the entire domain, which means the LCE of any predictor is just an approximation of the predictor’s overall calibration error. In contrast, DCE measures calibration error on disjoint two groups: x<0 and x>0. That is, DCE still measures the same quantity as before, enforcing that a predictor is both calibrated on x<0 and calibrated on x>0. It’s less clear whether DCE’s treatment of x>0 and x<0 as disjoint groups makes sense in this case where the clusters are nearly indistinguishable.
>
> **Q3: Computational Efficiency**
>
> > Is the proposed algorithm 2 computationally tractable?
>
> Multicalibration algorithms including Algorithm 2 are generally not computationally efficient. However, there’s been recent progress towards oracle-efficient algorithms for multicalibration, which our results would directly benefit from (e.g., [1]). It’s also worth noting that in practice multicalibration can be approximately implemented as efficient boosting algorithms (e.g. [2]); we see these potential extensions as exciting directions for future work.
>
> **On Gopalan et.al. 2021:**
>
> Thank you for the pointer. We will add an extended form of the following discussion to our revision: While Gopalan et al. 2021 also studies likelihood ratios, we take very different perspectives on how multicalibration relates to the likelihood ratio. In Gopalan et.al., the goal is to approximate the likelihood ratio as accurately as possible; their perspective is that multicalibration can be useful as a tool to relax pointwise to setwise accuracy. In our setting, our goal is not to approximate the likelihood ratios, but rather to make predictions that are multicalibrated with respect to the class of all plausible likelihoods consistent with our generative model.
>
> [1] Garg et.al. Oracle Efficient Online Multicalibration and Omniprediction
>
> [2] Globus-Harris et.al. Multicalibration as Boosting for Regression

---

> > ### Comment · Reviewer_iw9y · 2025-04-05
> >
> > I thank the authors for their detailed responses. I will keep my score!

---

### Official Review · Reviewer_DK2e · 2025-03-14

**Overall Recommendation:** 1

**Summary:**

This paper focuses on evaluating prediction performance on meaningful subpopulations rather than the overall population in a clustering problem. It proposes two levels of guarantees for capturing performance per subgroup: (1) evaluating the performance if assigning an individual to the most likely cluster, and (2) evaluating the performance if assigning each individual to all clusters weighted by their relative likelihood. The paper introduces a multi-objective algorithm to simultaneously handle both formalisms for all plausible underlying subpopulation structures, and evaluates the proposed algorithm in the context of online calibration as a case study.

**Claims And Evidence:**

This paper provides extensive theoretical proofs on different scenarios. However, the method is not evaluated properly empirically.

**Essential References Not Discussed:**

n/a

**Experimental Designs Or Analyses:**

This paper didn't provide any empirical studies nor experiments on real data.

**Methods And Evaluation Criteria:**

There are no evaluations on either simulated datasets or benchmark datasets.

**Other Comments Or Suggestions:**

n/a

**Other Strengths And Weaknesses:**

The paper is well-presented and the motivation is clearly presented.

However, the assumptions can limit its applicability to common scenarios in practice. There's no empirical justification of the method with simulated data or real data.

See Questions Section for more discussion.

**Questions For Authors:**

I have the following key concerns:

(1) I don't see any empirical studies nor experiments to justify your theory and your algorithm in both main text and supplementary. Can you provide empirical results to justify the effectiveness of your method?

(2) I wonder how much we can trust the underlying "naturally emerging" distributions. For example, in mixture models, we make assumptions about the structure of each subpopulation. However, these assumptions can often be incorrect, leading to problematic results [1]. Given the potential for model misspecification, would this method still hold? In comparison, the method proposed by [2] considers both subgroup clustering performance and model robustness when evaluating and making predictions. Could you provide some high-level comparison between the method proposed here versus the power posterior proposed in [2] from the methodology perspective?

(3) The framework of this paper is based on assumptions that may be challenging to scale to real-world applications. For instance, the paper assumes the data is generated from a mixture of distributions with a fixed number of subpopulations $k$. However, $k$ is typically unknown in clustering problems, and different choices of $k$ can significantly impact the structures and the number of subpopulations on which this paper relies.

[1] Cai et al (2020). Finite mixture models do not reliably learn the number of components. In Int. Conf. on Machine Learning, Online, 18–24 July 2021, pp. 1158–1169.

[2] Miller, J. W., & Dunson, D. B. (2019). Robust Bayesian inference via coarsening. Journal of the American Statistical Association.

**Relation To Broader Scientific Literature:**

n/a

**Theoretical Claims:**

I did not review every detail of the proofs. Please refer more to the other reviewers' comments regarding the correctness of the theory.

---

> ### Author Rebuttal · Authors · 2025-04-01
>
> Thanks for your review and questions!
>
> **Q1: On empirical results**
>
> > I don't see any empirical studies nor experiments to justify your theory and your algorithm in both main text and supplementary. Can you provide empirical results to justify the effectiveness of your method?
>
> We emphasize that the main contributions of our work are theoretical in nature. This paper contributes to the multicalibration literature which is a well-established field of theoretical research (e.g. [1-3]). The literature we draw from (such as learning algorithms and guarantees for Gaussian mixture models) are similarly in well-established theoretical fields (e.g. [4-5]). Given that our paper’s primary subject area is also Theory, we believe that a fully theoretical treatment is an appropriate and meaningful way to advance understanding and make foundational contributions to these theoretically rich fields.
>
> **Q2: On misspecification.**
>
> > Given the potential for model misspecification, would this method still hold?
>
> We believe that our multi-objective approach (Alg 2) is indeed useful for reducing the impact of misspecification of the generative model. This is because we never commit to learning a specific mixture distribution explicitly; instead, we are “robust” to any likelihood that could have been modeled by one's hypothesis class of clustering functions. Thus, even under misspecification, our method succeeds with provable guarantees as long as the “true” likelihood function is reasonably well-approximated by some function in a large empirical cover we construct.
>
> It is also important to note that the consequences of model misspecification in multicalibration are significantly more benign than misspecification in many other empirical settings. Multicalibration seeks to refine the average prediction accuracy of a classifier to hold also on a per-subgroup level. If the subpopulations are misspecified, the model still outputs a well-calibrated highly accurate predictor, but with a potentially coarsened per-subpopulation guarantee.
>
> > Could you provide high-level comparison between the method proposed here versus the power posterior proposed in Miller et.al. from the methodology perspective?
>
> Miller et al. studies robustness to uncertainty around identifying the clustering parameters of one’s data. In contrast, our goal is to make predictions that are high-quality across clusters; we are not focused on learning model parameters or densities directly. In fact, one of our paper’s main messages is that—for the purpose of providing subgroup guarantees—it is not necessary to identify the clustering parameters of your data. This is why we are able to get learning rates that are independent of cluster separation. That is, the robustness that Miller et al. studies is robustness that we show can be enjoyed entirely for free.
>
> Perhaps the methodological similarity between our multi-objective algorithm (Algorithm 2) and the “coarsening” idea in Miller can be understood as, essentially, improved performance arising from simultaneously considering many true underlying “likelihoods” — in our case, all possible likelihood functions consistent with the original function class, and in Miller et.al.’s, all possibilities within a perturbed neighborhood of a particular radius. We think a valuable direction for future work is to study whether methods for robustly learning likelihoods (such as Miller’s) could be useful as an alternative to our covering approach.
>
> **Q3: Knowing the number of clusters k.**
>
> > The paper assumes the data is generated from a mixture of distributions with a fixed number of subpopulations k. However, k  is typically unknown in clustering problems, and different choices of k can significantly impact the structures and the number of subpopulations on which this paper relies.
>
> While setting k is generally a sensitive parameter choice for clustering algorithms, it is *not* a sensitive parameter for our algorithm. Our algorithm doesn’t hinge on the data being exactly described by k clusters: instead, it provides guarantees for all plausible ways of clustering the data into *up to* k clusters. Thus, it suffices to just set k to be a generous upper bound on the number of clusters that you might expect in your data.
>
> All this said, we think that dealing with misspecification of the likelihood function class $\mathcal{F}$ and/or of $k$ would also be worthy avenues for study. As discussed above, we believe our multi-objective approach has potential in terms of providing such robustness guarantees, though we leave explicit development of these arguments to future work.
>
> [1] Hebert-Johnson et.al., Multicalibration: Calibration for the (computationally-identifiable) masses
>
> [2] Dwork et.al., Outcome Indistinguishability
>
> [3] Gopalan et.al., Ominpredictors
>
> [4] Azizyan et.al., Minimax theory for high-dimensional gaussian mixtures with sparse mean separation
>
> [5] Hardt and Price, Tight bounds for learning a mixture of two gaussians.

---

### Decision · Program_Chairs · 2025-05-01

**Decision:**

Accept (poster)

**Comment:**

The paper proposes a new perspective for multi-group guarantees in learning. A common viewpoint currently is to define groups as being identifiable by some (simple) function of the features. The paper suggests that groups can instead of identified by some natural clustering present in the feature space. An issue that arises then is that this clustering may not be known, and the paper shows that an approach which first clusters and then predicts gets a sub-optimal T^(2/3) rate on an online calibration task (and this might be a barrier for such direct unsupervised learning approaches). In contrast, the paper shows that T^(1/2) regret is possible, using a multicalibration algorithm on some cover over likelihood ratios.

Overall, I think the new statistical indistinguishability perspective on multi-group fairness (instead of defining it using a fixed function class), and also the results which get around barriers in unsupervised learning, are both interesting and relevant to the ICML community. Therefore, I recommend acceptance.